# Psychophysiological correlates of science communicators

**David Vagni[1], Gennaro Tartarisco[1], Simona Campisi[1], Loredana Cerbara[2], Marco Dedola[3], Alessandra Pedranghelu[4], Alexandra Castello[4], Francesca Gorini[4], Chiara Failla[1], Marco Tullio Liuzza[5], Antonio Tintori[2], Giovanni Pioggia[1], Marco Ferrazzoli[6,7‡*], Antonio Cerasa[8,9‡*]**

**1** Institute for biomedical research and innovation, National Research Council, IRIB-CNR, Messina, Italy, **2** Institute for Research on Population and Social Policies, National Research Council of Italy, Rome, Italy, **3** RAI-radiotelevisione italiana, Rome, Italy, **4** Press Office, National Research Council of Italy, Rome, Italy, **5** Department of Developmental Psychology and Socialization, Università di Padova, Padua, Italy, **6** National Research Council Piazzale Aldo Moro, Rome, Italy, **7** Tor Vergata University, Rome, Italy, **8** S. Anna Institute, Crotone, Italy, **9** Institute of BioImaging and Complex Biological Systems (IBSBC-CNR), Catanzaro, Italy

☢ These Authors equally contributed as first authors.
‡ MF and AC also equally contributed as corresponding authors.
* marco.ferrazzoli@cnr.it (MF); antonio.cerasa@cnr.it (AC)

## Abstract

We conducted a study in an ecological setting to evaluate the heart rate variability (HRV) of expert communicators during a live national primetime video interview. The study involved 32 expert science communicators, all with mid- to long-term experience in public speaking and outreach work, who were evaluated by an external jury to assess their communication skills. Prior to the experiment, participants completed an online survey to gather socio-demographic data, work-related information, and psychological profiles. The six indices of communication abilities assessed by jury were: Interest, Agreement, Engagement, Authoritativeness learning, and Clarity. HRV acquisitions were divided into three phases: baseline pre-interview, during the interview, and another baseline recording after the interview. Science communicators were characterized by high levels of self-esteem and prosociality, which were positively correlated with communication indices and inversely correlated with age. Evaluation of physiological responses showed that the total power and low-frequency components of HRV were significantly higher in the post-interview phase compared to both the interview and pre-interview phases. However, when we divided the entire group according to high and low Authoritativeness and Clarity indices, significant interactive effects were detected. Indeed, for the low Authoritativeness and Clarity subgroups, significant differences among all phases were observed, with total power decreasing from the pre-interview to the interview phase and increasing in the post-interview phase. This indicates a clear pattern of stress response and recovery. In contrast, the high Authoritativeness and Clarity subgroup showed less variation across phases, suggesting better stress regulation or less perceived stress during the interview. We provided the psychophysiological basis of science communication expertise that can affect the control of stress regulation during public speaking.

**Data availability statement:** All relevant data are within the manuscript and its Supporting Information files.

**Funding:** The author(s) received no specific funding for this work.

## 1. Introduction

Science communicators play a critical role in advocating for policy changes, correcting disinformation, and ensuring equitable access to knowledge [1]. As scientific topics often involve complex and nuanced information, the ability to convey this information clearly and engagingly is essential for fostering public understanding and trust in science [2]. A recent study by Houck and colleagues [3] used a preregistered, nationally representative survey in the United States to assess the impact of scientific expert communicators and messages. The study demonstrated that scientists have a larger influence on public opinion in areas where knowledge is based on non-ideological misperceptions than the identical science-based message from another source. Unfortunately, many topics are becoming ideologized, leading to the current crisis in scientific authority, underscoring the importance of effective science communication [4]. Scientific debate, traditionally viewed as authoritative and unquestioned, faces increasing challenges from anti-scientific movements and public mistrust. This crisis highlights the need for scientists to not only communicate their findings clearly but also to engage with the public in ways that are accessible and trustworthy. Improving scientific literacy and adapting communication strategies to address the public's concerns and misconceptions are crucial steps toward restoring faith in scientific authority.

Previous research has highlighted the importance of various skills and attributes in successful science communication. For instance, Maher et al. [5], proposed the three essential *A*ttributes (*A*s) of effective science communicators: *Accessible*, *Accountable* and *Adaptable*. Borowiec [6], instead, proposed ten simple rules for planning science communication activities: a) defining goals; b) figure out audience; c) picking an area of discussion; d) come up with a clear headline message; e) beware of jargon; f) show your audience why they should care; g) tell you audience how to react; h) get some feedback; i) consider equity and inclusion; l) the interdependence of the rules.

However, as it has already been shown in other specific works such as taxi drivers, mathematicians, chess players, noses, chefs, and sommeliers [7–10] the trained abilities of science communicators during high-stress situations, like live public broadcasts, may also reflect long-term expertise that induces brain specialization. Expertise development is not the same as simple experience. Most abilities, unlike riding a bike, never become highly skilled. Experts, on the other hand, reinvest cognitive capacity to further enhance performance [11]. Over the last two decades, a significant body of research in cognitive neuroscience has demonstrated the anatomical and functional variations in the brains of elite performers [12]. These studies have shown how sensory, motor, and cognitive processes interact to produce clear adaptive brain plasticity over the years-long process of skill learning. Understanding the neurophysiological correlates of science communicators can provide insights into how expertise modifies the neural substrates of task performance enabling them to manage stress and maintain effective communication under pressure.

Heart rate variability (HRV) refers to the variation in time between consecutive heartbeats and is a valuable indicator of the balance between the sympathetic and parasympathetic branches of the autonomic nervous system (ANS). The SNS is activated during stress, preparing the body for a "fight or flight" response by increasing heart rate and reducing HRV. On the other hand, the PNS helps mitigate this response by slowing the heart rate and increasing HRV [13]. However, during prolonged or intense stress, the PNS may not be able to adequately counterbalance the sympathetic activation, impairing the body's ability to cope with stress and leading to sustained reductions in HRV. HRV has also been linked with the activity of a flexible network of brain areas that are dynamically arranged in response to environmental stimuli [14]. Neuroimaging research suggests that HRV may be associated with a decreased sense of threat, mediated via cortical regions (such as the ventral side of the mPFC) involved

in the evaluation of stressful circumstances [15]. In the context of live communication, managing autonomic responses may be important, as it could influence aspects such as a communicator's performance, clarity, and overall effectiveness. Therefore, HRV patterns during high-level performance may serve as indicators of communication expertise.

In this experimental study, we investigated the HRV of science communicators during a live national video interview to explore how communication expertise impacts autonomic regulation. We hypothesized that greater expertise in science communication and perceived authority would be associated with better autonomic control, as compared to less experienced and less authoritative communicators. To test this hypothesis, we recorded HRV activity in a sample of researchers involved as science communicators with varying levels of scientific and communication experience, evaluated by an external jury.

## 2. Methods

### 2.1. Study enrollment: scientific communicators

The Press Office of the National Research Council (CNR) in Rome enrolled researchers with mid- to long-term experience in scientific communication between December 2, 2022 and April 4, 2024, taking into consideration only CNR workers. After expressing interest in the study, participants were asked to complete an online survey, which was used to screen for eligibility. Exclusion criteria were: a) less than 5 years of expertise in science communication events (including, for instance, public engagement, organization of cultural events, TV, radio, or newspaper interviews, and press releases); b) history of heart or pulmonary disease; c) being under pharmacological treatments for clinical pathologies affecting pressure or heart activity (i.e., diabetes; mood disorders).

From an initial cohort of 40 science communicators, thirty-two nonsmoking science communicators accepted to participate and were enrolled in the study (16 men age mean ± SD: 53.5 ± 8.6 years). None of the subjects had been taking any medications for at least 2 weeks before the study. On the day of the study, the subjects were instructed to avoid alcohol and caffeinated beverages for the preceding 12 hours and to abstain from heavy physical activity from the day before.

Written informed consent was obtained from all the volunteers, and the study was approved by the Ethical Committee Palermo 1 (report n. 8/2022–14/09/2022) and by the data protection officer of the CNR (N. 76260 del 20230314 2023-CNR0A00-007626).

### 2.2. Experimental setup

This sociological experiment was carried out in the RaiNews24 TV broadcast studios in Rome, Italy, during the "Scientists" program conducted by Marco Dedola. HRV was recorded during the live television interview (Fig 1). To maximize the ability to classify various forms of stress, subjects performed identical tasks under the same conditions. Specifically, interviews with participants for the TV show were scheduled on Wednesday between 7:40 PM and 8:00 PM. This procedure enabled a quantitative comparison between the performances of the science communicators based on both one-dimensional HRV metrics and the suggested method in two dimensions (see below).

### 2.3. Procedure

Participants arrived at the TV studios 30 to 60 minutes before the show. They were briefly informed about the interview and completed the final consent form. Fifteen minutes before they were directed to the restroom for the application of the Polar H10 chest band for heart rate variability (HRV) monitoring. This step aimed to mitigate the potential effects of bladder

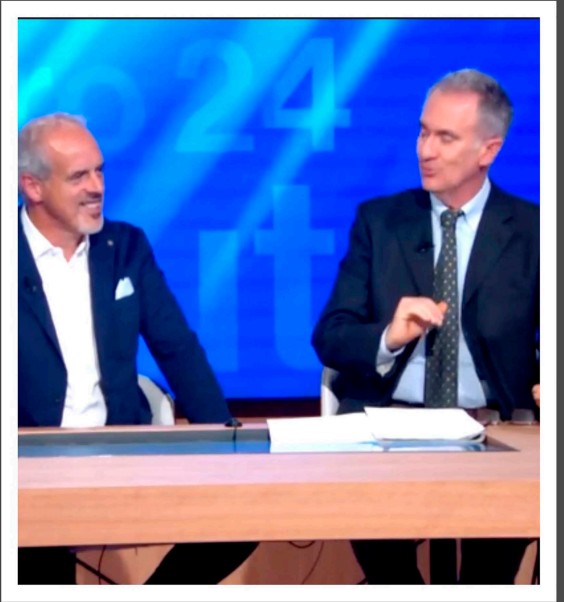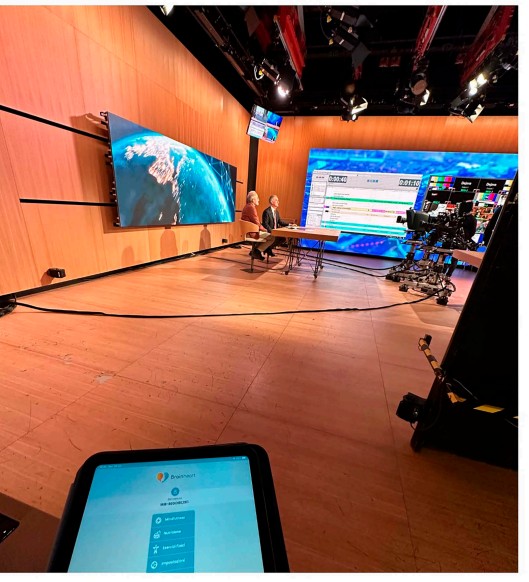

**Fig 1. Experimental Setup and live ECG recording during an on-air primetime TV interview: On the left side, the science communicator was interviewed by Marco Dedola, director of the "Scientists" TV show.** During the live Interview, communicators wore polar chests for HRV recording. Data was sent to the BrainHeart Platform running on a tablet (right side) located in the TV room. Reprinted from Marco Dedola and Andrea Dini under a CC BY license, with permission from PlosOne, original copyright 2025.

filling and gastric distension on HRV measurements [16]. Then, they were taken to a studio with a glass panel facing the studio, where they sat and chatted with the conductor while waiting for the broadcast. After the live interview, they went back to the first studio.

We designed an experiment to demonstrate how physical and mental states align with our two-dimensional representation. We specifically compared levels of mental stress during preparation and debriefing interviews versus actual interviews. Participants alternated between public exposure segments and private periods to establish a baseline level and monitor recovery periods before, between, and after the mental and emotional task.

The study employed a repeated-measures design with three conditions: Baseline1, Interview, and Baseline2. Specifically, the protocol was as follows:

- 5 minutes: sitting, preparing to go on air, informal chat about work-related topics;
- 10 minutes: sitting, live interview;
- 5 minutes: sitting, back in the first studio, debrief and informal chat about work-related topics.

The instructions for each stage were provided by one of the co-authors (D.V. or C.F.), who were present in the studio where the room temperature was maintained at 22 °C (71.6 °F). These instructions dictated the duration of the intervals for each session, ensuring a uniform procedure across all recording sessions and subjects. During the experiment, informal talking was only permitted during baseline and transition periods.

## 2.4. Socio-demographic and psychological measures

After providing written informed consent, the participants were invited to complete an online survey. The survey tool was based on the LimeSurvey community edition version 4.3 software

(www.limesurvey.org) supplied to the CNR. All the interviews were obtained immediately after the television interview. Personal data and health-related information recorded on the App were protected in terms of privacy following the last technologies developed in the field of mobile healthcare-related systems. The life cycle of the data was described in a Privacy Impact Assessment Document (DPIA) submitted for opinion to the Data Protection Officer (DPO). The entire survey takes less than two hours to complete. of the CNR.

After filling in an initial survey capturing social and demographic information, participants used the online survey system to fulfill work-related information and psychological questionnaires, including:

a) **Personality:** The HEXACO-60 [17] is a short personality inventory that measures the six major dimensions of personality: Honesty-Humility, Emotionality, Extraversion, Agreeableness, Conscientiousness, and Openness to Experience.

b) **Anxiety:** we are interested in evaluating only the state anxiety using the State-Trait Anxiety Inventory (STAI-Y Form Y-1) [18]. This scale had minimum and maximum scores of 20 and 80. 40–50 is the cutoff score for mild anxiety, 50–60 for moderate anxiety, and 60 and above for severe anxiety.

c) **Self-esteem:** The Rosenberg self-esteem scale [19] is one of the most widely used scales for measuring self-esteem. The total scores ranged from 0 to 40. The higher the score, presumably, the higher a person's self-esteem.

d) **Pro-Sociality:** The 16-item Prosociality Scale was created to evaluate individual variations in adults' inclinations to act altruistically [20]. The Prosociality Scale has been linked to emotional and empathic self-efficacy as well as agreeableness.

e) **Social desirability:** the Balanced Inventory of Desirable Responding (BIDR) short form (BIDR6, [21]) was used.

For all the scales, after recording the reverse items, we computed the summed score. Given the known shortcomings of Cronbach's alpha [22,23], which assumes a tau-equivalent (i.e., the same factor loadings for all the items), we computed the McDonald's total Omega [24] as an internal reliability index and considered acceptable values such as 0.70 or higher. For this purpose, we used the psych R package [25].

## 2.5. Evaluation of communication abilities

Twenty-six external students (21F, age range 19-63, mean age = 27.6, SD = 11.2) were recruited to assess the communication abilities of scientists. The enrollment took place at the University Tor Vergata (Rome, Italy) and the University "Magna Graecia" (Catanzaro, Italy). These students voluntarily offered their time to participate in the study. After completing the experimental study, all video interviews were downloaded from the Italian national television website (https://www.rainews.it/storie/futuro24). The order of the video was administered to each participant through a balanced Latin square procedure to control for sequence and order effects.

Participants were required to fill out a questionnaire where they assessed the communication ability of each scientist across six dimensions on a Likert-type response format:

- Clarity of Presentation, from 1 (*terrible*) to 5 (*excellent*)

- Learning Impact, from 1 (*no improvement*) to 5 (*notably*)

- Authoritativeness, from 1 (*not authoritative at all*) to 5 (*extremely authoritative*)

- Engagement of the episode, from 1 (*not interesting at all*) to 5 (*extremely interesting*)

- Interest in the topic, from 1 (*not interested at al*l) to 5 (*extremely interested*).

- Agreement with the scientist, from 1 (*not at all in agreement*) to 5 (*very much in agreement*)

The interviews were viewed throughout each of the multiple sessions that comprised the experiment. To reduce distractions, each participant viewed the video on their own in a controlled setting. Following each session, the completed surveys were gathered. To protect participant privacy and avoid any potential biases in evaluation, data were collected anonymously.

## 2.6. HRV apparatus

HRV activity was continuously recorded using the BrainHeart app [26]. This e-health platform is an innovative full-featured system that incorporates multiple parameters monitoring to improve users' lifestyles and, as a result, their psychophysical well-being. In this study, we utilized this application to record RR intervals and corresponding HRV parameters from the Polar H10 sensor chest strap (Polar Electro Oy, 2020 sampling rate: 1000Hz) [27].

**2.6.1. HRV preprocessing.** The RR time series represents the time intervals between successive R waves of the electrocardiogram (ECG) signal, reflecting the HRV. Accurate analysis of RR intervals provides reliable insights into ANS activity and psychophysiological stress response. However, RR time series often contain artifacts, which can significantly distort both time- and frequency-domain measurements, especially during prolonged HRV monitoring in naturalistic settings.

As an initial preprocessing step, we addressed these artifacts, including ectopic, extra, missed, and misaligned beats, using the robust automatic algorithm proposed by Lipponen and Tarvainen [28]. To ensure data integrity, we aimed to keep the percentage of corrected beats below 5% for each subject, as recommended by the Kubios HRV Software [29]. Following artifact removal, detrending was employed to remove slow nonstationary trends from the HRV signal. We utilized the smooth priors method introduced by Tarvainen [30], with a smoothing parameter of $\lambda = 500$. Finally, cubic spline interpolation was conducted at a rate of 4Hz to achieve evenly spaced samples suitable for frequency domain analysis [29]. Further details regarding the reprocessing steps are outlined in Fig 2. All preprocessing steps were performed in a Jupyter Notebook environment using Python 3.

**2.6.2. HRV postprocessing.** We evaluated variations in heart rate by computing various time-domain, frequency-domain, and non-linear measures. The preprocessed RR interval data from the Polar H10 device were analyzed using Kubios HRV Scientific Lite, a gold-standard HRV software [29]. Data for each participant, including gender and age, were imported into Kubios and segmented according to the different phases of the interview, as shown in Fig 3.

To account for the significant impact of period length on HRV features in both time-domain and frequency-domain analysis [31], we ensured that epochs were of the same length across all interview phases. Moreover, in line with the recommendations from the Task Force of the European Society of Cardiology and the North American Society of Pacing Electrophysiology, short-term recordings (~5 minutes) devoid of ectopy, missing data, and noise, were prioritized [32]. Consequently, 5-minute segments were predominantly selected, with 3-minute windows used when necessary. For each epoch, time-domain and frequency-domain measurements were selected based on the Task Force's guidelines. The HRV signal spectrum was derived from the interpolated discrete event series, representing the plot of $R_iR_{i-1}$ intervals against time, indicated at $R_i$ occurrence. The non-parametric Fast Fourier Transform method was employed, utilizing a window width of 300 seconds with 50% overlap. Mean HR and RR, as well as the standard deviation of the normal beat intervals (SDNN), were extracted for time

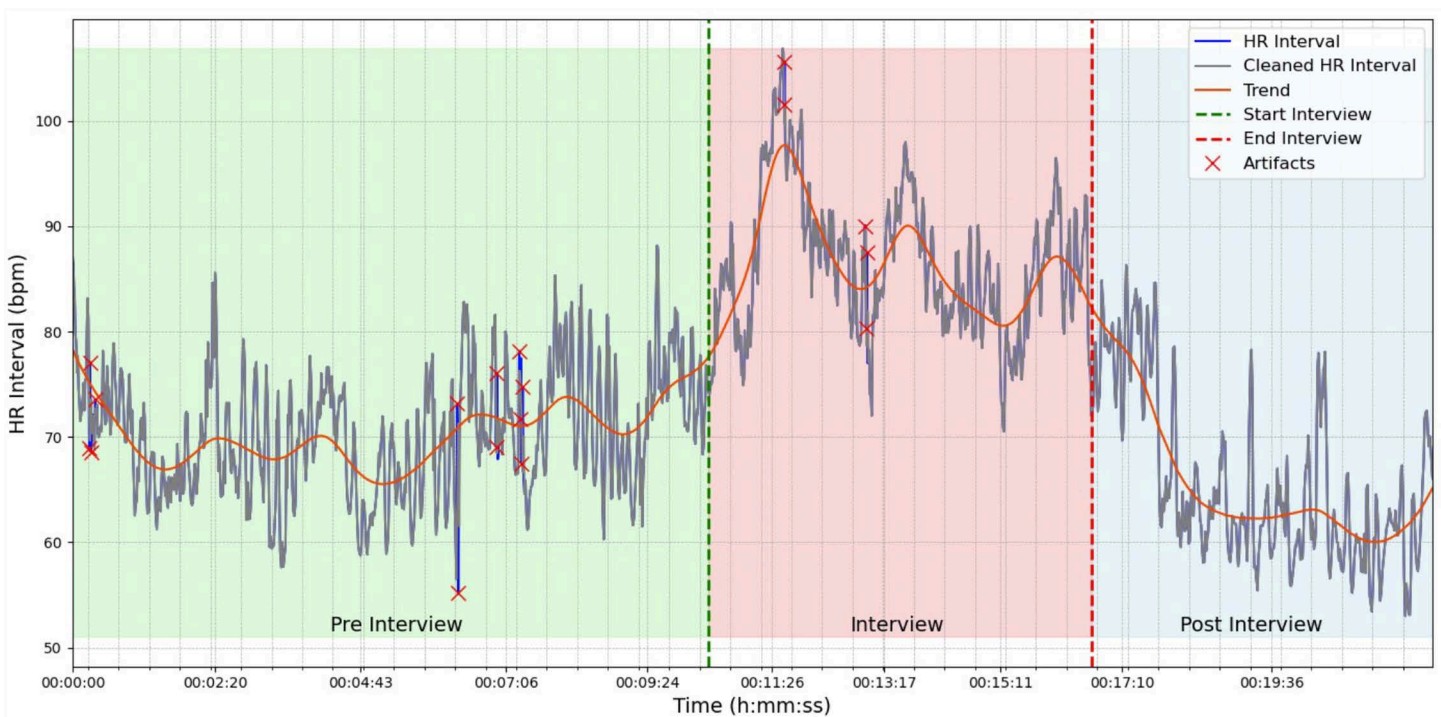

**Fig 2.  Heart rate dynamics during an interview assessment.: The figure shows the raw heart rate (HR) interval (blue line), the cleaned HR interval after artifact removal (gray line), and the trend obtained from the smooth priors method (orange line).** Artifacts, indicated by red crosses, were identified and interpolated using the automatic method proposed by Lipponen and Tarvainen (2019). The vertical dashed lines mark the start and end times of the interview, while the different phases of the interview process (pre-interview, during the interview, and post-interview) are indicated by the colored background areas.

domain HRV assessment. SDNN serves as a key time-domain measure of HRV and provides an overall assessment of autonomic variability, reflecting the capacity of the autonomic nervous system (ANS) to adapt to changing physiological demands [33]. Frequency-domain analyses focused on determining LF (low frequency, 0.04-0.15 Hz) and HF (high frequency, 0.15-0.4 Hz) powers in milliseconds squared to describe power distribution in spectral components. Additionally, the sum of the energy in the VLF, LF and HF bands, called Total Power, normalized HR (HRnu), and HF/LF ratio, were used as indicators for comparisons with previous studies. Furthermore, LF plus HF power (LHFP), and low-high frequency normalized difference (HLFND = LFnu - HFnu) were computed.

### 2.7.  *Normalized difference and bidimensional analysis*

The interpretation of HRV metrics has been extensively debated in the literature. HF is widely accepted as a marker of parasympathetic activity, reflecting respiratory sinus arrhythmia (RSA) [34]. In contrast, LF may be produced by both sympathetic and parasympathetic inputs, with evidence suggesting that it primarily reflects baroreflex-mediated autonomic modulation rather than direct sympathetic drive [35,36].

Additionally, the LF/HF ratio has been proposed as an index of sympathovagal balance but has faced significant criticism due to its dependence on multiple physiological factors and its susceptibility to distortions from non-linear relationships or extreme values [37].

To address these limitations, we implemented a two-dimensional analysis of HRV, examining the LF and HF components independently rather than relying on the LF/HF ratio. A ratio accurately reflects differences in the denominator only when the relationship between the numerator

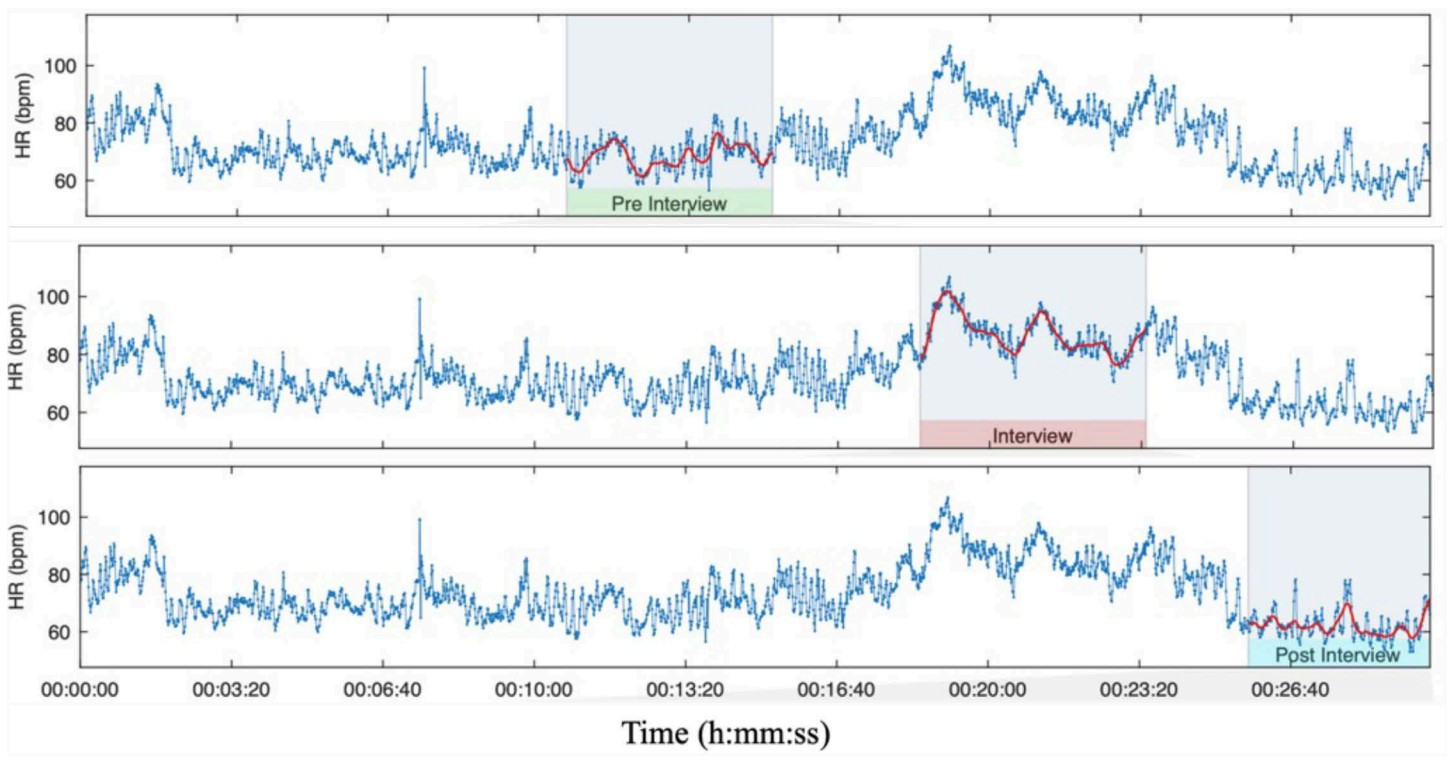

**Fig 3. Kubios output showing HR data segmented by interview phases.**

and denominator is linear and passes through the origin. If this condition isn't satisfied, the ratio can distort the true relationship, potentially leading to incorrect conclusions—either missing a real difference between groups or suggesting a difference that doesn't exist [38].

Furthermore, we introduced a new index, the Low-High Frequency Normalized Difference (LHFND), calculated as LHFND = (LF - HF)/ (LF + HF), which provides a bounded, symmetric measure (−1 to +1) of the relative dominance of LF and HF components. Unlike the LF/HF ratio, LHFND offers a more interpretable and balanced representation of automatic dynamics, mitigating distortions caused by extreme values. Although LHFND is a novel application in HRV research, its theoretical foundations are well supported by similar applications in other scientific domains, such as EEG signal processing [39] and hyperspectral imaging [40], where normalized differences provide consistent and interpretable measures of relative dominance. By applying this principle to HRV, LHFND captures subtle shifts in autonomic balance during dynamic conditions such as stress and recovery.

In addition, we combined LHFND with LHFP in a two-dimensional analysis, offering a novel perspective on autonomic balance. This approach enables us to capture subtle yet meaningful variations in autonomic dynamics that might be overlooked with traditional unidimensional metrics. Geometrically, LHFND corresponds to a normalized 45° rotation on the HF-LF plane, further enhancing interpretability in dynamic conditions such as stress and performance tasks.

## 2.8. Statistical analysis

The analyses were conducted using R [(33), version 4.4.0], R Studio [(34), version 1.2.1335], and SPSS [(34b), version 29.0.1.0]. To assess normal distribution, the Kolmogorov-Smirnov

test with Lilliefors significance correction and the Shapiro-Wilk test were employed. Outliers were identified as data points that lay beyond 1.5 times the interquartile range (IQR) from the first and third quartiles. Preliminary repeated measures ANOVAs were performed to examine the effects of the interview phase (pre-interview, interview, post-interview) on various heart rate variability (HRV) components, including Total Power, LF, HF, HFnu, RR, HR, SDNN, LF/HF, LHFND, and LHFP. Age and the logarithm of the number of publications were included as covariates, while gender was considered as a factor.

Additionally, because repeated-measures ANOVAs assume that the variances of the differences between all possible pairs of within-subject conditions are equal, we performed Mauchly's Test of Sphericity. A significant result on Mauchly's test signals a violation of sphericity, which can inflate Type I error rates. However, given the small sample size, we primarily utilized multivariate tests with Pillai's Trace for robustness.

For questionnaires evaluating communication abilities, mean scores and standard deviations were calculated for each dimension (Clarity, Learning, Authoritativeness, Engagement, Interest, and Agreement) to summarize the scientists' overall performance. Principal component analysis was conducted to assess the dimensionality, and paired t-tests were used to identify significant differences between dimensions. Correlation analyses explored the relationships between the different communication dimensions, and McDonald's Omega assessed the internal consistency of the questionnaire items for each dimension. Principal component analysis (PCA) with varimax rotation and Kaiser normalization was used to extract dimension factorization for the jury judgments.

A repeated measures ANOVA was then conducted to examine the effects of different judgment variables (Authoritativeness, Clarity, Engagement, Learning, Interest, Agreement), introducing them as covariates. To support this, we also performed multivariate ANOVAs using different HRV indices across the three phases as covariates (analyzing HF and LF together), with performance judgments as dependent variables. For multivariate analyses (e.g., MANOVA) with multiple outcome variables, we used Box's Test of Equality of Covariance Matrices to verify the assumption that each group has a similar covariance structure. If this test was found to be significant, indicating a violation of homogeneity, we employed corrections to mitigate its impact.

Therefore, communicators were categorized into high and low-performance groups using K-Means clustering based on the same judgment variables. Repeated measures ANOVAs were again conducted to examine the effects of the interview phase on HRV components (LF, HF, LHFND, LHFP, RR, HR, and SDNN), with age and the logarithm of publications as covariates, and performance clusters and gender as between-subjects factors. Additionally, we conducted clustering based on HF and LF values and compared differences between the groups on psychological, work-related, and performance variables, as well as their overlap with performance judgment clusters, using Chi-squared analysis.

Given the potential bidirectional relationship between the jury's judgments and HRV responses across different interview phases, this comprehensive approach allows us to explore both directions. By grouping participants according to performance judgments and HRV responses, we aim to identify key factors contributing to higher performance and understand how these factors correlate with demographic, personality, or scientific productivity dimensions.

## 2.9. Study registration

The study was approved by the Ethical Committee Palermo 1 (report n. 8/2022–14/09/2022) and by the Ethical Clearance Committee of the CNR (N. 76260 del 20230314 2023-CNR0A00-007626).

## 3. Results

### 3.1. Population characteristics

Work-related, and psychological characteristics are reported in **Table 1**. Our sample geographical distribution differs from the geographical distribution in Italy 72% work in central Italy. A large portion of our sample (87.5%) consists of leader researchers who coordinated groups for many years (41% from seven to ten years). Finally, 57% received at least one national/international scientific award.

Regarding dissemination activities, 34% use social media platforms daily to communicate science, 34% have completed communication courses, and 40% have held leadership positions in a social group (community, school, or university).

At a psychological level, the Internal reliability ranged from Omega total = .78 (HEXACO Open) to 0.94 (BIDR IM). Science communicators are characterized by normal levels of state

**Table 1. Work-related, and psychological characteristics of science communicators.**

| Work-related factors | | | |
|---|---|---|---|
| H-Index | 21.5 ± 14.7 | | |
| ERC-related membership | 59% PE | 37.5% LS | 3.5% SH |
| N° of international peer-reviewed publications | 12.5% more than 150 | 18.75% from 81-150 | 68.75% from 40 to 81 |
| N° of monographies | 37.5% from 4-10 | 62.5% from 1-3 | |
| N° invitations to scientific congress as speaker | 15.6% more than 100 | 43.8% from 30-100 | 40.6% from 10 to 30 |
| N° scientific projects as PI | 41% more than 20 | 50% from 6-20 | 9% from 1 to 5 |
| N° of professorships at university | 6.2% more than 10 | 22% from 6-10 | 71.8% from 0 to 5 |
| N° of training courses as a speaker (i.e., PhD; Master) | 56% more than 30 | 43.8% from 10-30 | 12.5% from 1 to 10 |
| Dissemination activities | | | |
| N° public speaking experiences (i.e., theater; coffee talk; festival; school) | 53% more than 30 | 31.4% from 10-30 | 15.6% from 1 to 10 |
| N° interviews in newspapers | 53% more than 30 | 22% from 10-30 | 25% from 1 to 10 |
| N° live television/radio interviews | 41% more than 20 | 24.7% from 6-20 | 34.3% from 1 to 5 |
| N° delayed television/radio interviews | 43% more than 20 | 32% from 6-20 | 25% from 1 to 5 |
| N° press releases | 50% more than 10 | 37.5% from 4-10 | 12.5% from 1 to 3 |
| Psychological data | | | |
| | Mean ± SD | Skewness | Kurtosis |
| STAI-Y1 | 33.5 ± 7.99 | 0.90 | 1.41 |
| Prosociality | 62.84 ± 8.03 | 0.42 | 1.42 |
| Self-esteem | 35.06 ± 4.05 | -0.59 | 0.72 |
| BIRD_SDE | 25.97 ± 4.22 | -0.16 | 0.75 |
| BIRD_IM | 30.44 ± 4.02 | -0.45 | 0.71 |
| HexacoOpen | 39.076 ± 4.96 | -0.34 | 0.88 |
| HexacoC | 37.84 ± 6.42 | -0.06 | 1.14 |
| HexacoA | 31.72 ± 6.04 | -0.94 | 1.34 |
| HexacoEx | 37.75 ± 4.01 | -0.19 | 0.71 |
| HexacoEm | 29.53 ± 5.48 | 0.36 | -0.57 |
| HexacoH | 42.19 ± 4.89 | -0.40 | -0.36 |

Abbreviations – STAI: State-Trait Anxiety Inventory; BIRD: Balanced Inventory of Desirable Responding. HexacoOpen: Openness to Experience; HexacoC: Conscientiousness; HexacoA: Agreeableness; HexacoEx: Extroversion; HexacoEm: Emotionality; HexacoH: Honesty

anxiety and high levels of self-esteem and prosociality (compared to previous normative data, see S1 file.docx). A t-test comparison revealed no gender differences in any psychological or work-related variables except for Emotionality, $t(30) = 2.387$, $p = 0.023$, with females scoring higher than males, $MD = 0.432 \pm 0.181$ and the number of publications, $t(30) = -2.300$, $p = 0.029$, with females scoring lower than males, $MD = -51.9 \pm 22.6$ (as well as h-index and the logarithm of those measures). Furthermore, an exact chi-square test of independence showed that there was a significant association between gender and public speaking training, $X^2$ (1, $N = 32$) = 6.788, $p = 0.023$, with more females (56.3%) than males (12.5%), who had the training.

## 3.2. Performance judgments by external jury

Two participants were eliminated from the original cohort because of a lack of video recording. The final group that was assessed for statistical analysis consisted of thirty people (see S1 Table).

The six dimensions that were assessed using the communication ability questionnaire all had normal distributions. Mean scores were positive, ranging from 3.27 for Interest in the Topic to 3.92 for Clarity of Presentation. Correlations among variables ranged from moderate to elevated.

For PCA, both *eigenvalues* and scree plot methods suggested a two-components solution, explaining 75% of the variance. After rotation, each component contributed almost equally. The first component encompassed variables related to the Content of the speech: Interest in the topic and Agreement with the communicator. The second component encompassed variables related to the Performance of the communicator: Authoritativeness, Clarity of presentation, and Learning impact. Engagement loaded nearly equally on both components. Full factor loadings are presented in Table 2 and represented in Fig 4.

We performed multiple correlations among demographic and psychological variables with Interview Judgments. Given the high number of correlations, $p$-values > 0.01 should be considered with caution. Age was correlated with Clarity, $r(30) = 0.442$, $p = 0.014$, Authoritativeness, $r(30) = 0.489$, $p = 0.006$, Engagement, $r(30) = 0.487$, $p = 0.006$, Interest, $r(30) = 0.418$, $p = 0.021$, Agreement, $r(30) = 0.375$, $p = 0.041$; Prosociality was correlated with Engagement, $r(30) = 0.590$, $p < 0.001$, Interest, $r(30) = 0.443$, $p = 0.014$, Agreement, $r(30) = 0.385$, $p = 0.036$; Self Esteem was correlated with Agreement, $r(30) = 0.453$, $p = 0.012$. Years teaching at university was correlated with Interest, $r(30) = 0.416$, $p = 0.022$; the number of printed interviews, number of live TV interviews, number of recorded TV interviews, and number of press communications were all correlated with Interest [0.354; 0.422] and more strongly correlated with Agreement [0.414; 0.583].

**Table 2. Descriptive Statistic and Principal Component Analysis of Interview Judgments.**

| Judgments | Descriptive Statistic | | | | Principal Component Analysis | | | |
|---|---|---|---|---|---|---|---|---|
| | Mean | S.E. | Full Interval | | Unrotated | | Rotated | |
| | – | – | Min | Max | Component-1 | Component-2 | Content | Performance |
| *Eigenvalue* | – | – | – | – | 3.404 | 1.120 | 2.288 | 2.235 |
| *Variance* | – | – | – | – | 0.567 | 0.187 | 0.381 | 0.373 |
| Interest | 3.269 | 0.050 | 2.770 | 3.810 | 0.789 | -0.542 | **0.943** | 0.164 |
| Agreement | 3.380 | 0.080 | 2.460 | 4.190 | 0.728 | -0.491 | **0.864** | 0.158 |
| Engagement | 3.456 | 0.046 | 2.920 | 4.000 | 0.879 | -0.125 | **0.716** | **0.525** |
| Authoritativeness | 3.499 | 0.043 | 2.880 | 4.080 | 0.660 | 0.507 | 0.118 | **0.823** |
| Learning | 3.680 | 0.052 | 3.040 | 4.190 | 0.675 | 0.444 | 0.172 | **0.790** |
| Clarity | 3.921 | 0.036 | 3.500 | 4.270 | 0.766 | 0.340 | 0.310 | **0.779** |

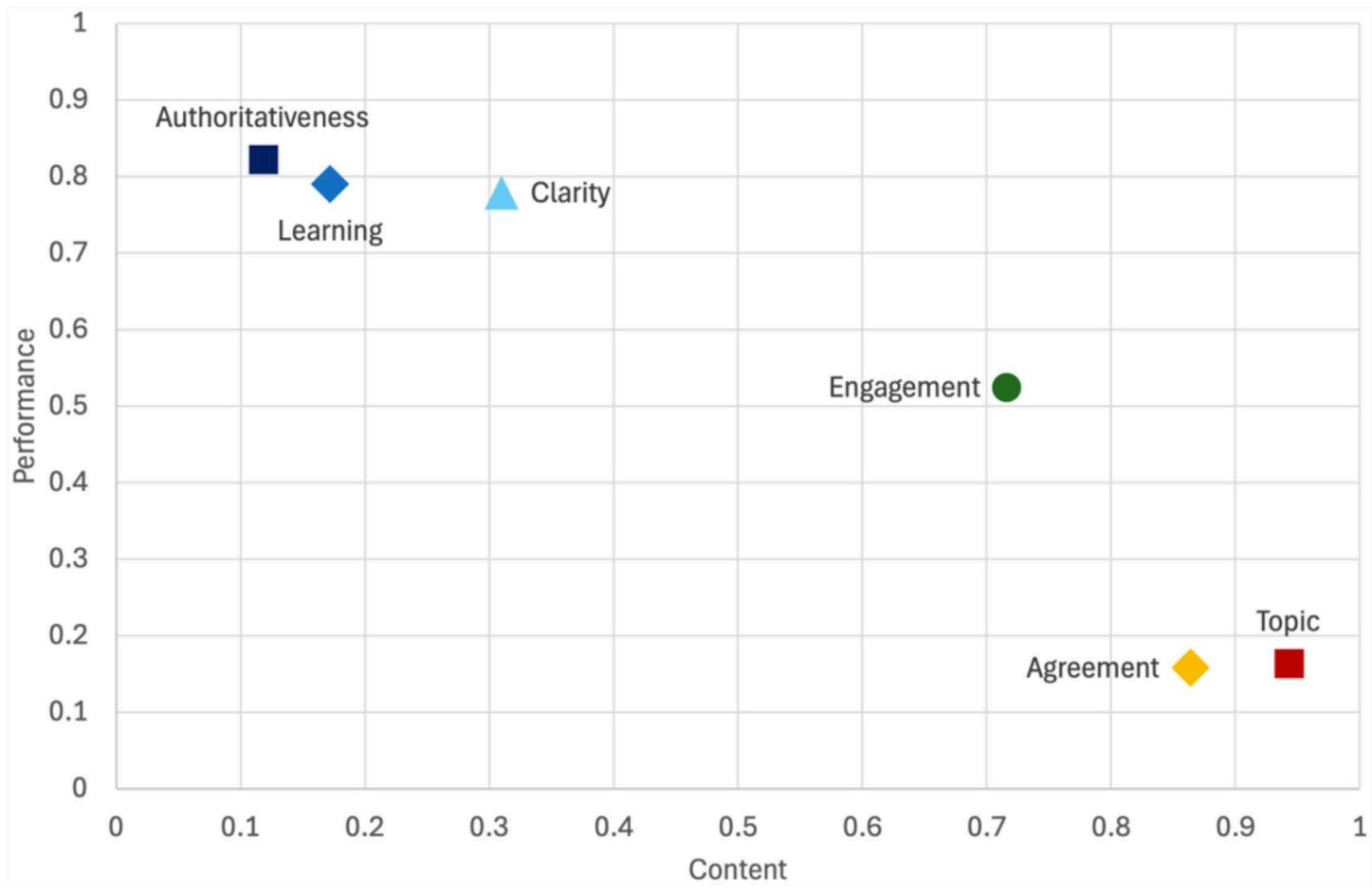

**Fig 4. Judgments loading on the two rotated Content and Performance components**

Log(pub) was negatively correlated with Authoritativeness, $r(30) = -0.474$, $p = 0.008$, the same hold for the raw number of publications and h-index. Log(pub) was positively correlated with the number of printed interviews, $r(30) = 0.384$, $p = 0.030$, and the number of live TV interviews, $r(30) = 0.428$, $p = 0.015$. $t$-test showed no significant effect of gender on judgment variables, while public speaking training had a positive effect on Authoritativeness, $t(28) = 2.314$, $p = 0.028$, and a negative relation with log(pub), $t(30) = -2.543$, $p = 0.016$.

Given the significant impact of Age and Log(pub) on judgment variables, we will include them as covariates in the main analysis to control for their influence and provide a clearer understanding of the primary relationships under investigation.

### 3.3. Heart rate variability indices across different phases

Seven individuals were eliminated from the original cohort because of elevated signal-to-noise recording during HRV acquisition. A final sample of 25 individuals was considered for these analyses (see S2 Table).

We performed multiple correlations among demographic and psychological variables with HRV indices. Given the high number of correlations, p-values > 0.01 should be considered with caution. Age was highly and negatively correlated with HF, LHFP, Total Power, and

SDNN during pre-interview and post-interview, r(25) < -0.46, $p$ = 0.02, and to a lesser degree with other HRV indices pre- and post-interview.

Pairwise comparisons, adjusted for age and Log(pub) as covariates, and factorized by Gender, revealed significant differences in physiological responses across the interview phases. Total power, LF, LHFP, and SDNN components of HRV were significantly higher in the post-interview phase compared to both the interview and pre-interview phases. HF components were significantly higher in the post-interview phase compared to the interview phase. RR intervals were significantly longer during both the post- and pre-interview phases compared to the interview phase, while the HR range was significantly lower during these phases. No significant interaction with phase was found for the LF/HR ratio, LHFND, and normalized HF.

When analyzing the data by gender, both males and females exhibited similar patterns. However, significant differences in total power, LHFP, LF, and HF between post-interview and interview phases were significant only for female participants, while a significant increase of SDNN from pre-interview to post-interview was present only in males. Female participants also had overall higher RR intervals and lower HR ranges compared to males. The detailed analysis is reported in Table 3.

### 3.4. HRV analyses with judgment variables as covariates

Repeated measures ANOVA was performed across the three experimental phases using Gender as a factor and age, log(pub) and the six judgment variables as covariates. Box's Test of Equality of Covariance Matrices and Mauchly's Test of Sphericity were non-significant for all the following analyses.

When LF was used as dependent variable, multivariate analysis led to a significant main effect of Phase, $F(2,14)$ = 5.266, $p$ = 0.020, $\eta_p^2$ = 0.429, and a non-significant interaction of Phase and Authority, $F(2,14)$ = 5.266, $p$ = 0.064, $\eta_p^2$ = 0.324, with a significant within-subject effect, $F(2,38)$ = 4.315, $p$ = 0.023, $\eta_p^2$ = 0.233. No significant between-subjects effect was found. There was a significant difference between the post-interview phase and both the pre-interview and interview phases with an increase in post-interview from interview, MD = 648 ± 146, $p$ = 0.001, and from pre-interview, MD = 455 ± 126, $p$ = 0.008. Within-Subjects Contrast of Phase x Authority interaction was significant for post-interview against interview phase, $p$ = 0.036, with higher Authority decreasing LF post-interview from the interview phase.

Using HF as a dependent variable, there was a significant interaction effect between Phase and Clarity, $F(2,14)$ = 4.613, $p$ = 0.029, $\eta_p^2$ = 0.397. Within-Subjects Contrast of Phase x Authority interaction was significant for post-interview against interview phase, $p$ = 0.019, with higher Clarity increasing HF post-interview from the interview phase.

When LHFP was used as dependent variable, multivariate analysis led to a significant main effect of Phase, $F(2,14)$ = 5.266, $p$ = 0.020, $\eta_p^2$ = 0.429, and a non-significant interaction of Phase and Authority, $F(2,14)$ = 2.846, $p$ = 0.092, $\eta_p^2$ = 0.289, with a significant within-subject effect, $F(2,38)$ = 3.385, $p$ = 0.047, $\eta_p^2$ = 0.184. No significant between-subjects effect was found. There was a significant difference between the post-interview phase and both the pre-interview and interview phases with an increase in post-interview from interview, MD = 789 ± 171, $p$ = 0.001, and from pre-interview, MD = 531 ± 155, $p$ = 0.011.

Using LHFND as a dependent variable, multivariate analysis led to a non-significant interaction of Phase and Learning, $F(2,14)$ = 3.284, $p$ = 0.068, $\eta_p^2$ = 0.319, with a significant within-subject effect, $F(2,38)$ = 3.368, $p$ = 0.048, $\eta_p^2$ = 0.183. A barely significant between-subject effect was found for Age, $F(1,15)$ = 4.488, $p$ = 0.051, $\eta_p^2$ = 0.230. Within-Subjects Contrast of Phase x Learning-interaction was significant for the pre-interview against interview phase, $p$ = 0.030, with higher Learning decreasing LHFND during interview from the pre-interview phase.

**Table 3. Selected Heart Rate Variability (HRV) measures during each 5-min segment in the pre-interview, interview, and post-interview phases.**

| Group | Time Segment | Pre-interview mean (95% C.I) | Interview mean (95% C.I) | Post-interview mean (95% CI) | p-value | Effect | Gender difference (p-value) |
|---|---|---|---|---|---|---|---|
| Total (N = 25) | Total power (ms²) | 985 (676, 1294) | 730 (432, 1029) | 1519 (1088, 1949) | 0.002 | POST> (INT; PRE) | 0.400 |
| | LF (ms²) | 643 (419, 868) | 449 (258, 640) | 1088 (722, 1455) | 0.003 | POST> (INT; PRE) | 0.227 |
| | HF (ms²) | 295 (177, 413) | 232 (120, 343) | 373 (272, 474) | 0.005 | POST> INT | 0.769 |
| | HF (nu) | 31 (25,36) | 32 (28,37) | 28 (23,34) | 0.52 | – | 0.152 |
| | RR interval (ms) | 680 (644, 717) | 586 (555, 617) | 708 (669, 746) | <0.001 | (POST; PRE)> INT | 0.004 (Female> Male) |
| | HR Range (bpm) | 90 (85, 95) | 105 (99, 110) | 87 (82, 91) | <0.001 | (POST; PRE) < INT | 0.004 (Female < Male) |
| | SDNN (ms) | 29.3 (24.8, 33.7) | 24.6 (19.6, 29.8) | 36.0 (31.1, 40.8) | <0.001 | POST> (INT; PRE) | 0.288 |
| | LF/HF ratio | 3.21 (2.16, 4.26) | 2.71 (2.02, 3.41) | 3.66 (2.16, 5.16) | 0.455 | – | 0.411 |
| | LHFP (ms²) | 938 (630, 1247) | 680 (395, 966) | 1462 (1037, 1886) | 0.001 | POST> (INT; PRE) | 0.422 |
| | LHFND | 0.384 (0.278, 0.490) | 0.358 (0.264, 0.452) | 0.431 (0.324, 0.537) | 0.536 | – | 0.156 |
| Female (N = 14) | Total power (ms²) | 1168 (711, 1626) | 800 (357, 1243) | 1705 (1067, 2345) | 0.018 | POST>INT | |
| | LF (ms²) | 874 (542, 1207) | 519 (235, 802) | 1259 (716, 1802) | 0.035 | POST>INT | |
| | HF (ms²) | 240 (65, 415) | 222 (57, 388) | 384 (235, 533) | 0.018 | POST>INT | |
| | HF (nu) | 26 (18,33) | 29 (22,36) | 27 (19,35) | 0.737 | – | |
| | RR interval (ms) | 746 (691, 780) | 640 (594, 686) | 782 (725, 840) | <0.001 | (POST; PRE)> INT | |
| | HR Range (bpm) | 82 (75, 89) | 95 (90, 103) | 78 (71, 85) | <0.001 | (POST; PRE) < INT | |
| | SDNN (ms) | 32.6 (26.0, 39.2) | 27.5 (19.9, 35.1) | 37.8 (30.7, 44.9) | 0.021 | POST>INT | |
| | LF/HF ratio | 3.26 (2.80, 5.92) | 2.80 (1.78, 3.83) | 3.73 (1.50, 5.95) | 0.181 | – | |
| | LHFP (ms²) | 1114 (657, 1572) | 741 (318, 1165) | 1643 (1013, 2272) | 0.014 | POST>INT | |
| | LHFND | 0.490 (0.333, 0.646) | 0.421 (0.282, 0.561) | 0.457 (0.299, 0.651) | 0.738 | – | |
| Male (N = 11) | Total power (ms²) | 802 (271, 1332) | 661 (147, 1174) | 1331 (590, 2072) | 0.099 | – | |
| | LF (ms²) | 412 (27, 798) | 379 (50, 708) | 918 (288, 1548) | 0.084 | – | |
| | HF (ms²) | 350 (147, 553) | 241 (49, 433) | 362 (189, 535) | 0.187 | – | |
| | HF (nu) | 36 (27,45) | 36 (28,44) | 30 (21,39) | 0.434 | – | |
| | RR interval (ms) | 615 (552, 680) | 532 (479, 585) | 633 (567, 700) | <0.001 | (POST; PRE)> INT | |
| | HR Range (bpm) | 98 (90, 106) | 114 (105, 124) | 95 (88, 103) | <0.001 | (POST; PRE) < INT | |
| | SDNN (ms) | 25.9 (18.3, 33.5) | 21.8 (13.0, 30.6) | 34.1 (25.8, 42.4) | 0.008 | POST> (INT; PRE) | |
| | LF/HF ratio | 2.06 (0.25, 3.87) | 2.67 (1.43, 3.82) | 3.59 (1.01, 6.17) | 0.289 | – | |
| | LHFP (ms²) | 763 (232, 1293) | 620 (129, 1111) | 1280 (550, 2010) | 0.090 | – | |
| | LHFND | 0.279 (0.098, 0.460) | 0.294 (0.132, 0.456) | 0.404 (0.221, 0.587) | 0.436 | – | |

**Legends:** *Covariates appearing in the model are evaluated at the following values: Age = 52.64, log(pub) = 3.9231. C.I. (confidence interval); p-value denotes paired sample t-test from corresponding pre-performance measurement. F (ms²): low frequency (millisecond squared). HF (ms²): high frequency (millisecond squared). nu: normalized units. RR interval: beat-beat interval. HR (bpm): heart rate (beats per minute), SDNN: Standard Deviation of NN intervals, LHFP (bpm): LF + HF (ms²), LHFND (nu): LF (nu) - HF (nu),. POST: baseline post-interview phase; PRE: baseline pre-interview phase. INT: Interview phase*

When HR was used as dependent variable, the multivariate effect of Phase was not significant, but we found only a between-subject effect of Engagement, $F(1,15) = 5.477$, $p = 0.034$, $\eta_p^2 = 0.267$, and Gender, $F(1,15) = 8.213$, $p = 0.012$, $\eta_p^2 = 0.354$. Both female gender and higher engagement reduce HR.

Using RR as a dependent variable, multivariate analysis led to a significant main effect of Phase, $F(2,14) = 6.592$, $p = 0.010$, $\eta_p^2 = 0.319$. An almost significant between-subject effect was found for Clarity, $F(1,15) = 4.265$, $p = 0.057$, $\eta_p^2 = 0.221$, and a significant effect for Engagement, $F(1,15) = 4.957$, $p = 0.042$, $\eta_p^2 = 0.248$, and Gender, $F(1,15) = 9.521$, $p = 0.008$, $\eta_p^2 = 0.388$, females have higher RR, while participants with higher RR have higher Engagement but possibly lower Clarity. Within-subjects contrast of Phase found lower RR during interview compared with pre-interview, MD = -94 ± 9, $p < 0.001$, and post-interview, MD = -121 ± 11, $p < 0.001$.

A summary table of statistically significant findings is reported in Table 4.

**Table 4. Statistically Significant Results for Heart Rate Variability (HRV) Indices with Judgment Variables as Covariates.**

| DV | Effect | Analysis | $F$(df1, df2) | $p$ | $\eta_p^2$ | Notes |
|---|---|---|---|---|---|---|
| **LF (ms²)** | Phase | Multivariate | 5.266 (2,14) | 0,020 | 0,429 | Post-interview> Interview [MD = 648 ± 146, p = 0.001]; Post-interview> Pre-interview [MD = 455 ± 126, p = 0.008]. |
| | Phase × Authority | Within-Subjects | 4.315 (2,38) | 0,023 | 0,233 | Higher Authority decreased LF from interview to post-interview (p = 0.036). |
| **HF (ms²)** | Phase × Clarity | Multivariate | 4.613 (2,14) | 0,029 | 0,397 | Greater Clarity increased HF from interview to post-interview [Contrast p = 0.019]. |
| **LHFP (ms²)** | Phase | Multivariate | 5.266 (2,14) | 0,020 | 0,429 | Post-interview> Interview [MD = 789 ± 171, p = 0.001]; Post-interview> Pre-interview [MD = 531 ± 155, p = 0.011]. |
| **LHFND** | Phase | Within-Subjects | 3.368 (2,38) | 0,048 | 0,183 | Significant effect across phases. |
| **HR (bpm)** | Engagement | Between-Subjects | 5.477 (1,15) | 0,034 | 0,267 | Higher Engagement is associated with lower HR. |
| | Gender | Between-Subjects | 8.213 (1,15) | 0,012 | 0,354 | Female participants have lower HR. |
| **RR (ms)** | Phase | Within-Subjects | 6.592 (2,14) | 0,010 | 0,319 | Interview < Pre-interview [MD = −94 ± 9, p < 0.001]; Interview < Post-interview [MD = −121 ± 11, p < 0.001]. |
| | Engagement | Between-Subjects | 4.957 (1,15) | 0,042 | 0,248 | Higher Engagement relates to higher RR. |
| | Gender | Between-Subjects | 9.521 (1,15) | 0,008 | 0,388 | Female participants show higher RR. |

**LF**: Low-frequency power of HRV; **HF**: High-frequency power of HRV; **LHFP**: Sum of LF and HF powers; **LHFND**: Normalized difference between LF and HF; **HR**: Heart rate; **RR**: Interval between R peaks. **DV (Dependent Variable).** The physiological measure being analyzed (e.g., LF, HF, LHFP, etc.). **Analysis.** Indicates whether the effect is (a) within-subject (e.g., Phase × AuthorityB) or (b) between-subject (e.g., Age, Gender, ClarityB). **F (df1, df2).** The F-statistic and relevant degrees of freedom. **p.** The exact p-value reported (or threshold if only given as "< 0.001"). **ηp².** Partial eta squared, a measure of effect size.

**Notes.** Key interpretation points: whether the measure increased or decreased in a given phase or group, pairwise comparisons (MD, mean difference). All F-values, degrees of freedom (df1, df2), p-values, partial eta-squared (ηp²) values, and mean differences (MD) refer to the specific contrasts or effects reported in the manuscript. Some contrasts are reported only in text (e.g., "Within-Subjects Contrast") where the primary ANOVA yielded significance, and post-hoc or planned contrasts further clarified the direction of the effect. Non-significant interactions or effects (e.g., p ≥ 0.05) are omitted for clarity.

## 3.5. Communication performance clustering

Communicators were divided into high (HA) and low Authoritativeness (LA) groups leading to 16 HA and 14 LA participants, with cluster center $C_{HA}$ = 3.66 and $C_{LA}$ = 3.31, F(1,28) = 34.06, $p$ < 0.001. Clustering for Clarity led to $C_{HC}$ = 4.05 and $C_{LA}$ = 3.73, F(1,28) = 51.52, $p$ < 0.001, with 18 and 12 communicators each, for Engagement, $C_{HE}$ = 3.63 and $C_{LE}$ = 3.24, F(1,28) = 42.59, $p$ < 0.001, with 17 and 13 communicators each, for Learning, $C_{HL}$ = 3.83 and $C_{LL}$ = 3.33, F(1,28) = 52.51, $p$ < 0.001, with 21 and 9 communicators each. Factors based on judgment clusters have been indicated with a subscript "B" (binary) to distinguish them from the continuous variable. For additional data see S1 file.docx

**3.5.1.** ***Heart rate variability analysis and authority.*** When using Authority$_B$ to cluster participants in repeated measures ANOVA with LF as a dependent variable, Box's Test of Equality of Covariance Matrices was significant, $p$ = 0.009 but Mauchly's Test of Sphericity was non-significant. Multivariate tests revealed no significant main effects of phase, gender, or HRV or their interaction with Age and Log(pub). However, there was a significant interaction between Authority$_B$ and Phase, $F$(2,18) = 4.171, $p$ = 0.032, $\eta_p^2$ = 0.317. No significant between-subjects effect was found. Multivariate tests of the interaction showed a significant effect of Phase in the LA group, $F$(2,18) = 10.611, $p$ < 0.001, $\eta_p^2$ = 0.541, but not in the HA group. In the LA group there was a significant difference among all Phases with a decrease of LF from pre-interview phase to interview, MD = -692 ± 220, $p$ = 0.016, and an increase in post-interview, MD = 1344 ± 285, $p$ < 0.001 (Fig 5).

In repeated measures ANOVA with HF as a dependent variable, Box's Test of Equality of Covariance Matrices and Mauchly's Test of Sphericity were non-significant. Multivariate tests revealed no significant effects. However, there was a significant between-subjects effect of age, $F$(1,19) = 6.157, $p$ = 0.016, $\eta_p^2$ = 0.245, with HF decreasing in all phases while the age increases.

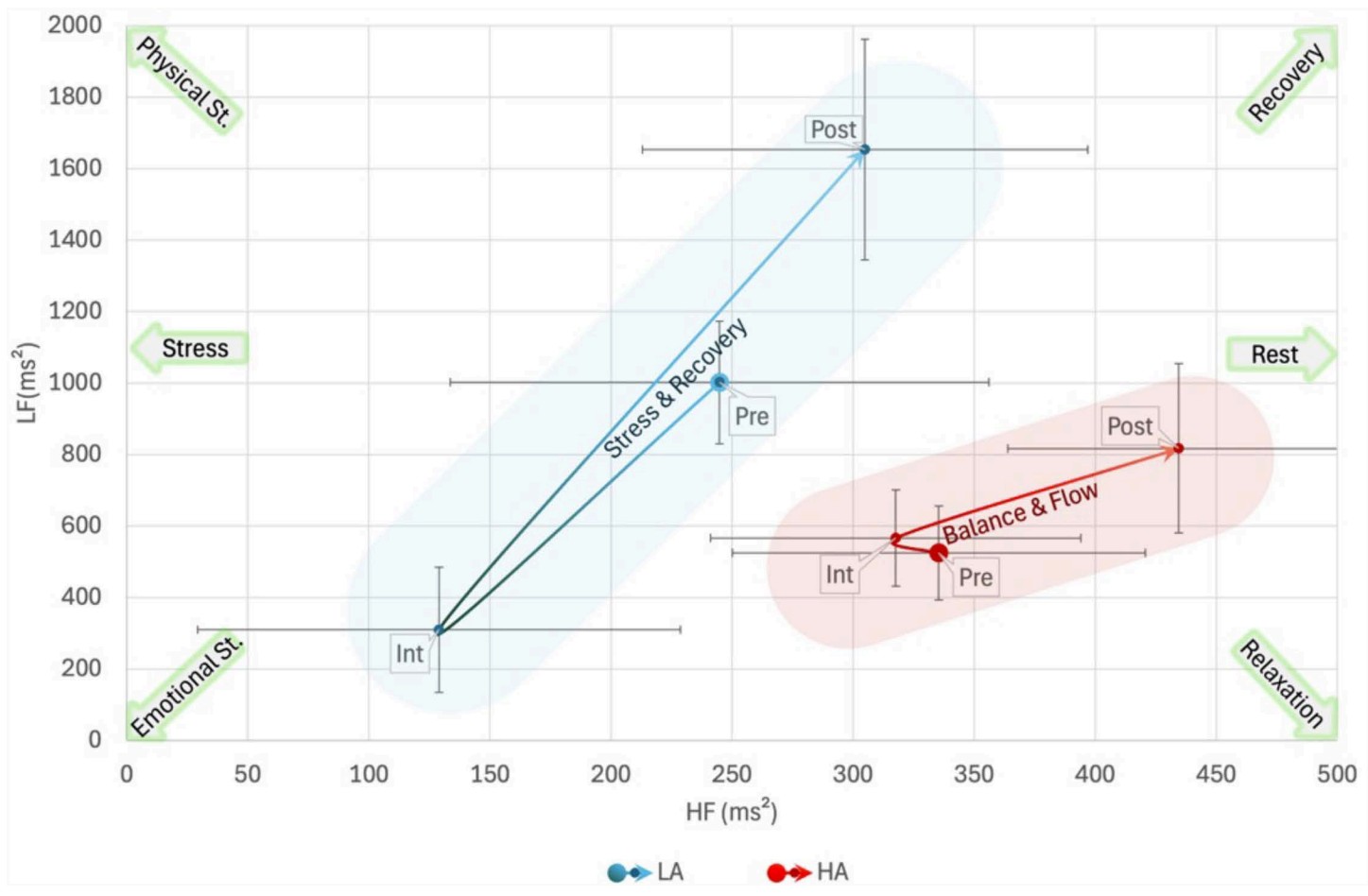

**Fig 5. Bidimensional High-Low Frequency (HF, LF) plane representation of the interaction among Phase and High-Low Authoritativeness Groups: In the Low Authority (LA) group there was a significant effect among all the three phases, moving from accumulated stress to acute stress and finally beginning recovery.** Otherwise, in the High Authority (HA) group, all the three phases were characterized by a balanced and "flow" condition of moderate activation. Shaded areas are for representational purposes only. Pre: baseline activity pre-interview; Int: Interview; Post: baseline activity post-interview.

In LHFP analysis, Box's Test of Equality of Covariance Matrices was significant, $p = 0.009$ but Mauchly's Test of Sphericity was non-significant. Multivariate tests revealed only a barely significant interaction between Authority$_B$ and Phase, $F(2,18) = 3.513$, $p = 0.052$, $\eta_p^2 = 0.281$, but the significance of lower bound on the within-subject effect was $p = 0.049$, and the Greenhouse-Geisser correction led to a $p = 0.020$ for the effect. No significant between-subjects effect was found. Multivariate tests of the interaction showed a significant effect of Phase in the LA group, $F(2,18) = 10.806$, $p < 0.001$, $\eta_p^2 = 0.546$, but not in the HA group. In the LA group there was a significant difference among all Phases with a decrease of LHFP from pre-interview phase to interview, MD $= -808 \pm 290$, $p = 0.035$, and an increase in post-interview, MD $= 1520 \pm 318$, $p < 0.001$.

In LHFND analysis, Box's Test of Equality of Covariance Matrices and Mauchly's Test of Sphericity were non-significant. Multivariate tests revealed no significant effects of phase or HRV or their interaction with Age and Log(pub). However, there was a significant between-subjects effect of Authority$_B$, $F(1,19) = 5.358$, $p = 0.032$, $\eta_p^2 = 0.220$, with MD $= 0.192 \pm 0.82$, toward a more balanced LHFND in the HA group (Fig 6).

In RR analysis, Box's Test of Equality of Covariance Matrices was significant, $p = 0.039$, and Mauchly's Test of Sphericity was non-significant. Multivariate tests revealed a significant

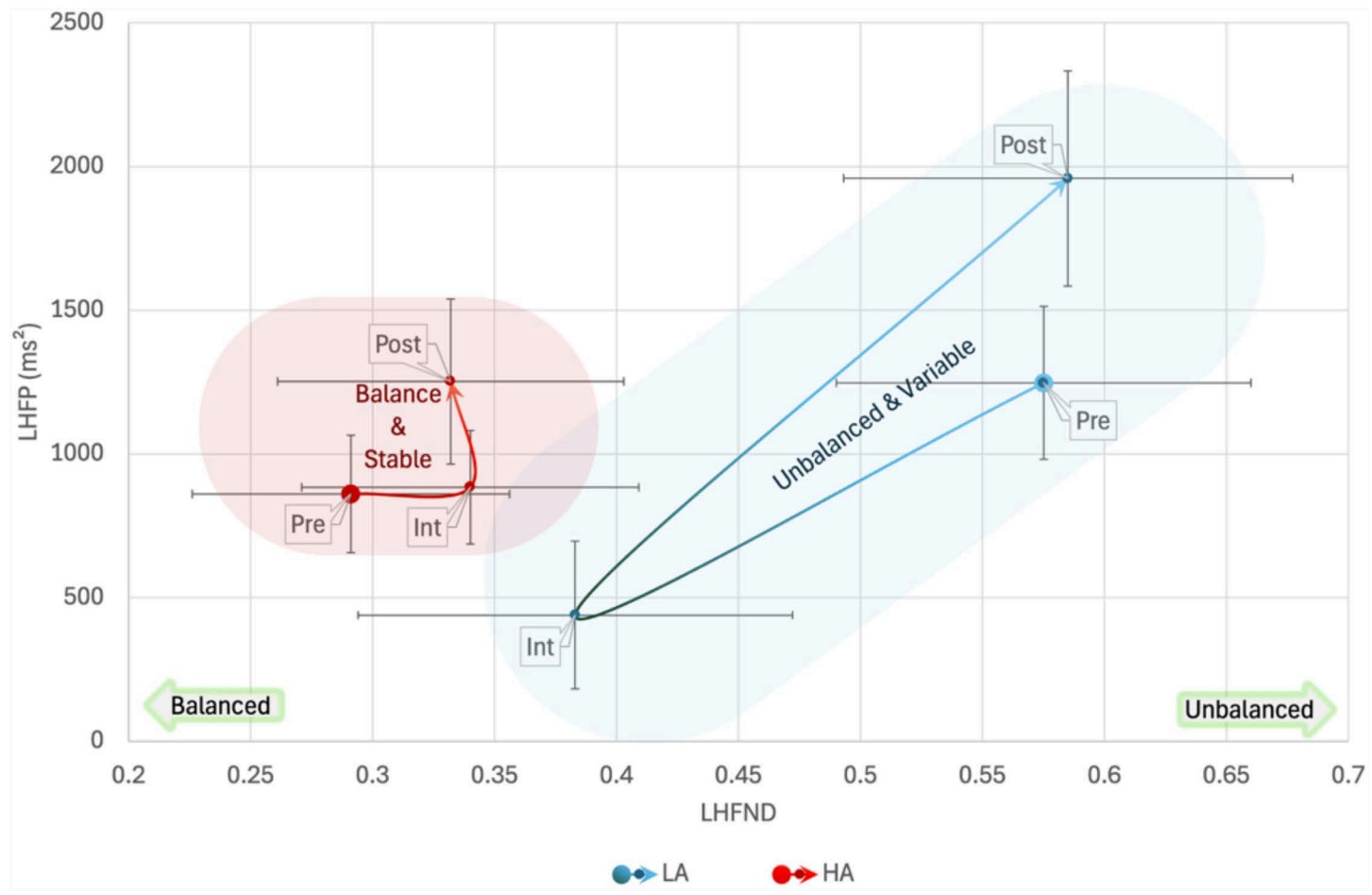

**Fig 6. Bidimensional Low-High Frequency Normalized Difference and High-Low Frequency Power (LHFND, LHFP) plane representation of the interaction among Phase and High-Low Authoritativeness Groups. In the Low Authority (LA) group there was a higher variability in power (LHFP) with a strong decrease during the interview and a lower overall balance than in the High Authority (HA) group.** Shaded areas are for representational purposes only. Pre: baseline activity pre-interview; Int: Interview; Post: baseline activity post-interview.

interaction between Phase and Authority$_B$. Furthermore, there was a significant between-subjects effect of Gender, $F(1,19) = 9.905$, $p = 0.005$, $\eta_p^2 = 0.343$, with MD = 124 ± 40, toward a larger RR in females, and an almost significant effect of Age, $F(1,19) = 4.192$, $p = 0.055$, $\eta_p^2 = 0.181$, with an increase in RR with age. Pairwise comparison among groups revealed a significant difference during the pre-interview phase, with LA having larger RR intervals than HA, MD = 92 ± 42, $p = 0.044$. Furthermore, there was a significant decrease in RR during the interview phase compared to the pre- and post-interview phases, $p < 0.001$, even if the decrease during the interview phase was more intense in the LA group compared to the HA group (Fig 7).

In HR analysis, Box's Test of Equality of Covariance Matrices was significant, $p = 0.008$, and Mauchly's Test of Sphericity was non-significant. Multivariate tests revealed a non-significant effect of Phase, $F(2,18) = 3.276$, $p = 0.061$, $\eta_p^2 = 0.267$, with a significant within-subject effect, $F(2,38) = 3.276$, $p = 0.026$, $\eta_p^2 = 0.175$. Furthermore, there was a significant between-subjects effect of Gender, $F(1,19) = 9.489$, $p = 0.006$, $\eta_p^2 = 0.333$, with MD = -17.0 ± 3.4, toward a smaller SD in females, and a significant effect of Age, $F(1,19) = 5.653$, $p = 0.028$, $\eta_p^2 = 0.229$, with a decrease in HR with age. Pairwise comparison among different phases

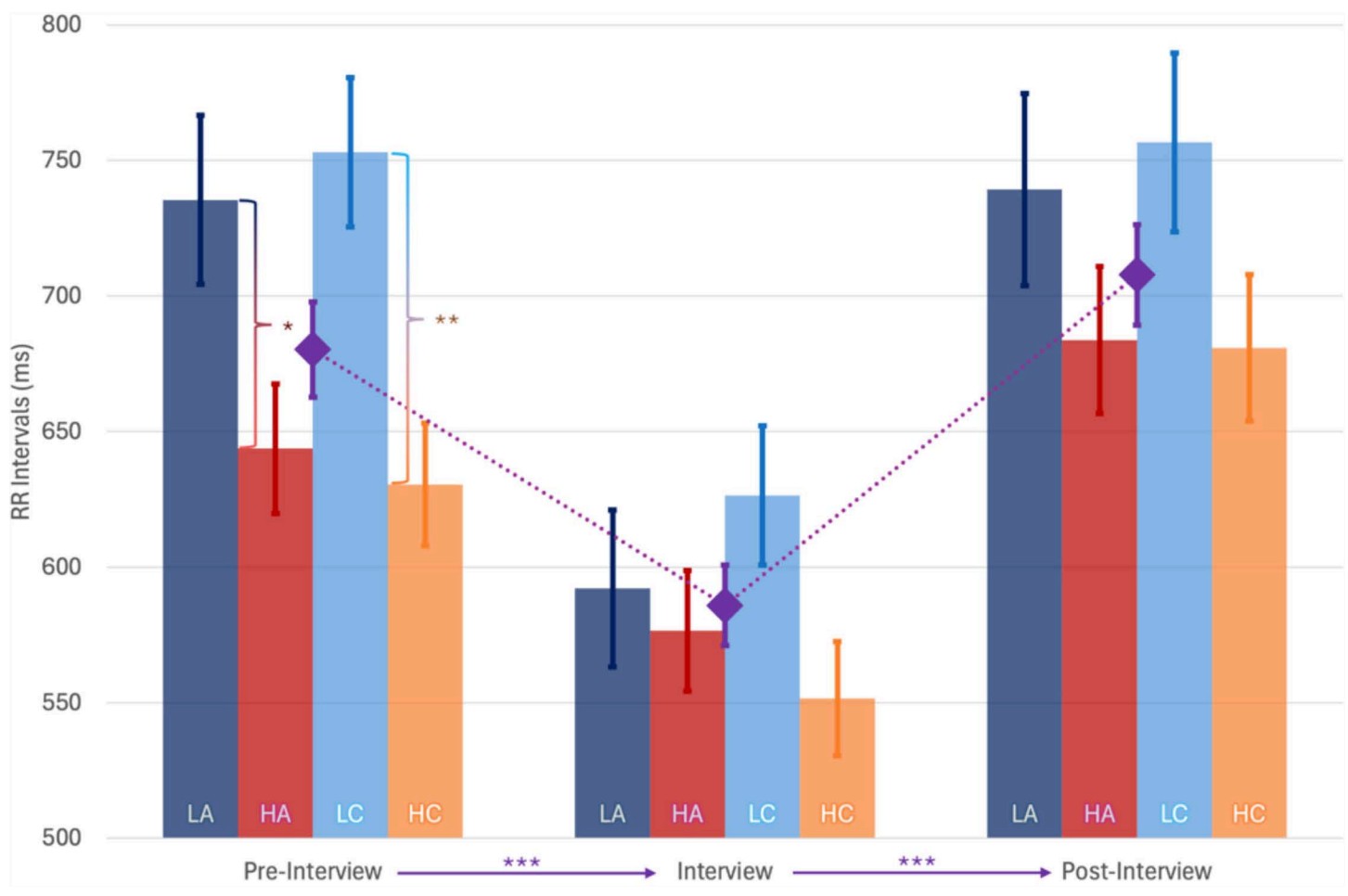

**Fig 7. Average Time Between Successive R-wave Peaks (RR Intervals) across pre-interview, interview and post-interview phases for High and Low Clarity and Authority groups: RR decreases from the pre-interview to the interview phase and increases again in the post-interview.** In the High Authority (HA) and High Clarity (HC) groups, RR pre-interview was lower than in the Low Authority (LA) and Low Clarity (LC) groups, respectively. During the interview, the LC group has a slightly higher RR than the HC group. * **p** < 0.05, ** **p** < 0.01, *** **p** ≤ 0.001. The diamond is the value for the whole group.

revealed an increase in HR during the interview phase, compared to pre-interview, MD = 15.9 ± 1.6, $p < 0.001$, and post-interview, MD = 18.4 ± 2.1, $p < 0.001$ (Fig 8).

A summary table of statistically significant findings is reported in Table 5.

**3.5.2  *Heart rate variability analysis and clarity.*** When using $Clarity_B$ to cluster participants in repeated measures ANOVA with LF as a dependent variable, Box's Test of Equality of Covariance Matrices was significant, $p = 0.048$ but Mauchly's Test of Sphericity was non-significant. Multivariate tests revealed a significant interaction between $Clarity_{yB}$ and Phase, $F(2,18) = 5.462$, $p = 0.014$, $\eta_p^2 = 0.378$, and a significant between-subjects effect of $Clarity_B$, $F(1,19) = 7.306$, $p = 0.014$, $\eta_p^2 = 0.278$. Between subject contrast, revealed a difference of MD = 605 ± 224, between LC and HC. Multivariate test of the interaction showed a significant effect of Phase in the LA group, $F(2,18) = 14.082$, $p < 0.001$, $\eta_p^2 = 0.610$, but not in the HA group. In the LA group there was a significant difference among all Phases with a decrease of LF from pre-interview phase to interview, MD = -662 ± 221, $p = 0.022$, and an increase in post-interview, MD = 1406 ± 258, $p < 0.001$. Furthermore, there was a significant difference between LC and HC groups in pre-interview phase, MD = 697 ± 234, $F(1,19)$ =

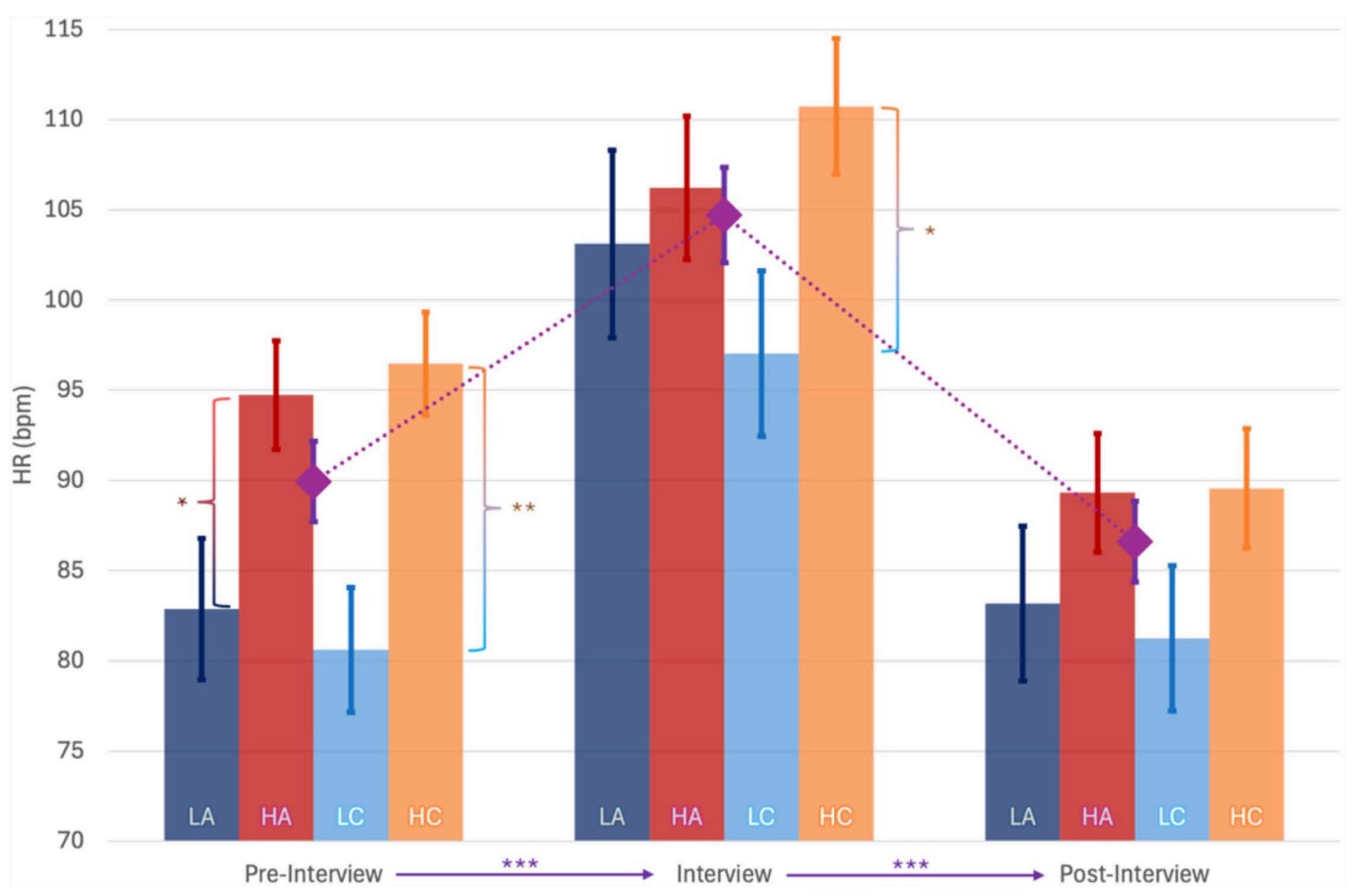

**Fig 8. Average Heart Rate across pre-interview, interview and post-interview phases for High and Low Clarity and Authority groups: Heart Rate (HR) increases from pre-interview to interview phase and decreases again in post-interview.** In the High Clarity group (HC), HR was higher than in the Low Clarity group (LC). LA: Low Authority group, HA, High Authority Group. * **p** < 0.05, ** **p** < 0.01, *** **p** ≤ 0.001. The diamond is the value for the whole group.

8.893, $p = 0.008$, $\eta_p^2 = 0.319$, and post-interview phase, MD = 1176 ± 379, $F(1,19) = 9.638$, $p = 0.006$, $\eta_p^2 = 0.337$, but not during the interview (Fig 9).

In repeated measures ANOVA with HF as a dependent variable, Box's Test of Equality of Covariance Matrices was significant, $p = 0.010$ but Mauchly's Test of Sphericity was non-significant. No significant within- or between-subject effects were found.

In LHFP analysis, Box's Test of Equality of Covariance Matrices was significant, $p = 0.018$ but Mauchly's Test of Sphericity was non-significant. Multivariate tests revealed a significant interaction between Clarity$_B$ and Phase, $F(2,18) = 4.078$, $p = 0.035$, $\eta_p^2 = 0.312$, and a significant between-subjects effect of Clarity$_B$, $F(1,19) = 5.799$, $p = 0.026$, $\eta_p^2 = 0.234$. Between subject contrast, revealed a difference of MD = 774 ± 321, between LC and HC. Multivariate test of the interaction showed a significant effect of Phase in the LA group, $F(2,18) = 12.626$, $p < 0.001$, $\eta_p^2 = 0.584$, but not in the HC group. In the LC group there was a significant difference among all Phases with a decrease of LF from pre-interview phase to interview, MD = -790 ± 285, $p = 0.036$, and an increase in post-interview, MD = 1545 ± 300, $p < 0.001$. Furthermore, there was a significant difference between LC and HC groups in pre-interview phase, MD = 946 ± 340, $F(1,19) = 7.736$, $p = 0.012$, $\eta_p^2 = 0.289$,

**Table 5. Statistically Significant Results for Heart Rate Variability (HRV) Indices for High and Low Authority Groups.**

| DV | Effect | Analysis | F(df1, df2) | p | $\eta_p^2$ | Notes |
|---|---|---|---|---|---|---|
| **LF (ms²)** | Authority$_B$ × Phase | Multivariate | 4.171 (2,18) | 0,032 | 0,317 | - LA group: significant phase effects |
| | Phase effect in LA group | Within-Subjects | 10.611 (2,18) | <0.001 | 0,541 | Pre-Interview ↓ Interview (p = 0.016), Interview ↑ Post-Interview (p < 0.001) |
| **HF (ms²)** | Age | Between-Subjects | 6.157 (1,19) | 0,016 | 0,245 | HF decreases with age |
| **LHFP (ms²)** | Authority$_B$ × Phase | Multivariate | 3.513 (2,18) | 0.052 | 0,281 | Corrected p < 0.05 (Greenhouse-Geisser p = 0.020) |
| | Phase effect in LA group | Pairwise Comparisons | 10.806 (2,18) | <0.001 | 0,546 | Pre-Interview ↓ Interview (p = 0.035), Interview ↑ Post-Interview (p < 0.001) |
| **LHFND** | Authority$_B$ | Between-Subjects | 5.358 (1,19) | 0,032 | 0,220 | More balanced LHFND in HA group (MD = 0.192 ± 0.82) |
| **HR (bpm)** | Phase | Within-Subjects | 3.276 (2,38) | 0,026 | 0,175 | HR increases during interview phase |
| | Gender | Between-Subjects | 9.489 (1,19) | 0,006 | 0,333 | Lower HR in females (MD = -17.0 ± 3.4) |
| | Age | Between-Subjects | 5.653 (1,19) | 0,028 | 0,229 | HR decreases with increasing age |
| **RR (ms)** | Authority$_B$ × Phase | Multivariate | 6.315 (2,18) | 0,008 | 0,412 | Significant interaction with greater decrease in RR in LA vs. HA |
| | Phase comparisons | Pairwise Comparisons | 76.867 (2,18) | <0.001 | 0,895 | Interview < Pre-Interview (MD = -113 ± 12) & Interview < Post-Interview (MD = -146 ± 20) |
| | Authority$_B$ in Pre-Interview | Pairwise Comparisons | 4.668 (1,19) | 0,044 | 0,197 | LA > HA (MD = 92 ± 42) |
| | Gender | Between-Subjects | 9.905 (1,19) | 0,005 | 0,343 | Larger RR in females (MD = 124 ± 40) |

˙DV (Dependent Variable). The physiological measure being analyzed (e.g., LF, HF, LHFP, etc.). LF: Low-frequency power of HRV; HF: High-frequency power of HRV; LHFP: Sum of LF and HF powers; LHFND: Normalized difference between LF and HF; HR: Heart rate; RR: Interval between R peaks. Analysis. Indicates whether the effect is (a) within-subject (e.g., Phase × Authority$_B$) or (b) between-subject (e.g., Age, Gender, Clarity$_B$). F (df1, df2). The F-statistic and relevant degrees of freedom. p. The exact p-value reported (or threshold if only given as "< 0.001"). $\eta p^2$. Partial eta squared, a measure of effect size. Notes. Key interpretation points: whether the measure increased or decreased in each phase or group, pairwise comparisons (MD, mean difference), etc. All F-values, degrees of freedom (df1, df2), p-values, partial eta-squared ($\eta p^2$) values, and mean differences (MD) refer to the specific contrasts or effects reported in the manuscript. Some contrasts are reported only in text (e.g., "Within-Subjects Contrast") where the primary ANOVA yielded significance, post-hoc or planned contrasts further clarified the direction of the effect. Non-significant interactions or effects (e.g., p ≥ 0.05) are omitted for clarity.

and post-interview phase, MD = 1305 ± 462, $F(1,19) = 7.997$, $p = 0.011$, $\eta_p^2 = 0.296$, but not during the interview (Fig 10).

In LHFND analysis, Box's Test of Equality of Covariance Matrices and Mauchly's Test of Sphericity were non-significant. Multivariate tests revealed no significant effects. However, there was a significant between-subjects effect of log(pub), $F(1,19) = 6.846$, $p = 0.017$, $\eta_p^2 = 0.265$, toward a less balanced LHFND with the increase in number of publications.

In RR analysis, no significant multivariate effects were observed. Nevertheless, we found between-subjects effect for Age, $F(1,19) = 7.848$, $p = 0.011$, $\eta_p^2 = 0.292$, log(pub), $F(1,19) = 6.017$, $p = 0.024$, $\eta_p^2 = 0.241$, Gender, $F(1,19) = 14.48$, $p = 0.001$, $\eta_p^2 = 0.433$, and Clarity$_B$, $F(1,19) = 5.791$, $p = 0.026$, $\eta_p^2 = 0.234$ (Fig 7). Older age, female gender, higher number of publications and lower clarity, increase RR.

In HR analysis, Box's Test of Equality of Covariance Matrices was significant, $p = 0.005$ but Mauchly's Test of Sphericity was non-significant. Multivariate tests revealed a significant main effect of Phase, $F(2,18) = 4.290$, $p = 0.030$, $\eta_p^2 = 0.323$, and a significant between-subjects effect of Clarity$_B$, $F(1,19) = 5.812$, $p = 0.026$, $\eta_p^2 = 0.234$, Age, $F(1,19) = 10.17$, $p = 0.005$, $\eta_p^2 = 0.348$, log(pub), $F(1,19) = 5.428$, $p = 0.031$, $\eta_p^2 = 0.222$, Gender, $F(1,19) = 13.97$, $p = 0.001$, $\eta_p^2 = 0.424$. HR decreases being female, older, having less publications and in the lower clarity group. Pairwise comparison among different phases revealed an increase in HR during the interview phase, compared to pre-interview, MD = 15.3 ± 1.7, $p < 0.001$, and post-interview, MD = 18.5 ± 1.8, $p < 0.001$ (Fig 8).

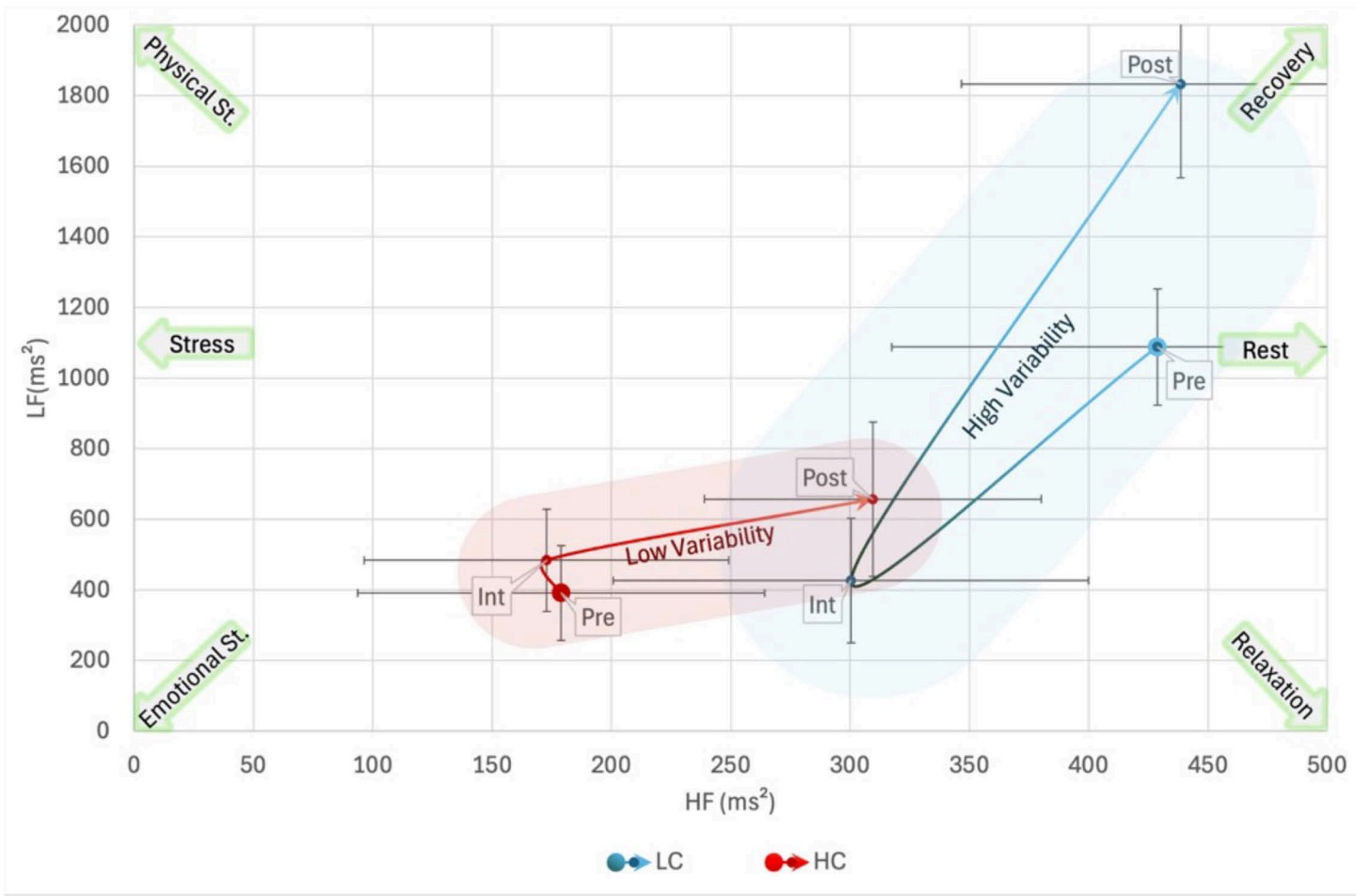

**Fig 9. Bidimensional High-Low Frequency (HF, LF) plane representation of the interaction among Phase and High-Low Clarity Groups. In the Low Clarity (LC) group there was a large variability in the low frequencies (LF).** Otherwise, in the High Clarity (HC) group, low-frequency variability was lower and stable throughout the phases while there was lower power in the high frequencies (HF) with similar pre-interview and interview values, pointing to an increased parasympathetic inhibition. Shaded areas are for representational purposes only. Pre: baseline activity pre-interview; Int: Interview; Post: baseline activity post-interview.

A summary table of statistically significant findings is reported in Table 6.

## 4. Discussion

To become an efficient communicator several cognitive and emotional abilities need to be acquired and maintained for a long time to manage stress and maintain effective communication under pressure. In this study, we provide the first psychophysiological basis of expertise in science communicators.

### 4.1. Psychological, work, and demographic profile of science communicators

At a psychological level, our findings revealed that science communicators generally exhibit normal levels of state anxiety, but high levels of self-esteem and prosocial behavior. This profile suggests that science communicators are well-equipped psychologically to handle the demands of their roles, which often involve public speaking and engagement in potentially stressful situations.

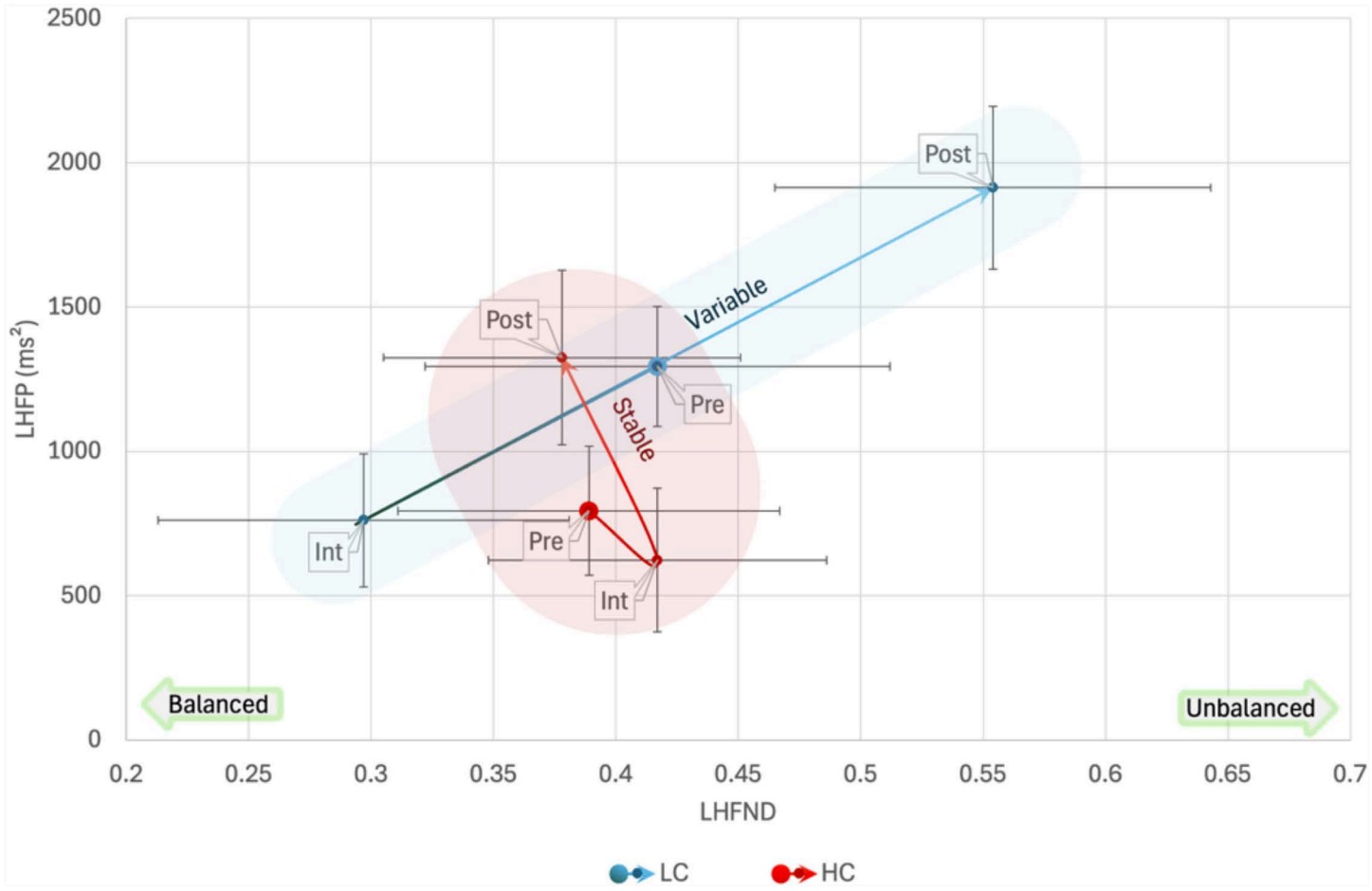

**Fig 10. Bidimensional Low-High Frequency Normalized Difference and High-Low Frequency Power (LHFND, LHFP) plane representation of the interaction among Phase and High-Low Clarity Groups. In the Low Clarity (LC) there was a high variability in both low-high frequency power and in the balance between high frequencies (HF) and low frequencies (LF).** Otherwise, in the High Clarity (HC) group, both features remain more stable across the different phases. Shaded areas are for representational purposes only.

We did not find significant gender differences in most psychological and work-related variables, except for emotionality, publication metrics. Women reported higher levels of emotionality compared to men, which aligns with existing literature [41]. This trait has been shown as the one where the greatest effect size was observed both in a college (Cohen's $d = 1.23$) and a community sample (Cohen's $d = 0.96$) ensuring the validation of the HEXACO-60 on Canadian samples [42]. More importantly, such a difference has since emerged in every culture, as a cross-country study by Lee & Ashton [43] showed differences spanning from Cohen's $d = 0.48$ (Indonesia) to Cohen's $d = 1.14$ (Netherlands), with an average Cohen's $d = 0.84$, which is large. This finding might reflect gender differences in emotional expressiveness or in how emotional experiences are internalized and reported, even though the literature on gender differences in personality seems to rule out that these differences are due to patriarchal culture since they appear to be larger, not smaller, in more gender-equal countries [44]. Higher emotionality could potentially influence how female science communicators experience and manage stress during public engagements. However, we found no gender difference in the judged performance or analyzed HRV indices.

**Table 6. Statistically Significant Results for Heart Rate Variability (HRV) Indices for High and Low Clarity Groups.**

| DV | Effect | Analysis | $F$(df1, df2) | $p$ | $\eta_p^2$ | Notes |
|---|---|---|---|---|---|---|
| **LF (ms²)** | Clarity$_B$ × Phase | Multivariate | 5.462 (2,18) | 0,014 | 0,378 | Significant phase effects in LC group |
| | Phase effect in LC group | Within-Subjects | 14.082 (2,18) | <0.001 | 0,610 | Pre-Interview ↓ Interview (p = 0.022), Interview ↑ Post-Interview (p < 0.001) |
| | Clarity$_B$ | Between-Subjects | 7.306 (1,19) | 0,014 | 0,278 | LC> HC (MD = 605 ± 224) |
| | LC vs. HC (Pre-Interview) | Pairwise Comparisons | 8.893 (1,19) | 0,008 | 0,319 | LC> HC (MD = 697 ± 234) |
| **LHFP (ms²)** | Clarity$_B$ × Phase | Multivariate | 4.078 (2,18) | 0,035 | 0,312 | Significant phase effects in LC group |
| | Phase effect in LC group | Pairwise Comparisons | 12.626 (2,18) | <0.001 | 0,584 | Pre-Interview ↓ Interview (p = 0.036), Interview ↑ Post-Interview (p < 0.001) |
| | Clarity$_B$ | Between-Subjects | 5.799 (1,19) | 0,026 | 0,234 | LC> HC (MD = 774 ± 321) |
| | LC vs. HC (Pre-Interview) | Pairwise Comparisons | 7.736 (1,19) | 0,012 | 0,289 | LC> HC (MD = 946 ± 340) |
| **LHFND** | log(pub) | Between-Subjects | 6.846 (1,19) | 0,017 | 0,265 | More publications → less balanced LHFND |
| **HR (bpm)** | Phase | Multivariate | 4.290 (2,18) | 0,030 | 0,323 | HR increases during interview phase |
| | Gender | Between-Subjects | 13.97 (1,19) | 0,001 | 0,424 | Females → lower HR |
| | Age | Between-Subjects | 10.17 (1,19) | 0,005 | 0,348 | HR decreases with increasing age |
| **RR (ms)** | Gender | Between-Subjects | 14.48 (1,19) | 0,001 | 0,433 | Females → larger RR |
| | Age | Between-Subjects | 7.848 (1,19) | 0,011 | 0,292 | Older participants → larger RR |

• **LF**: Low-frequency power of HRV; **HF**: High-frequency power of HRV; **LHFP**: Sum of LF and HF powers; **LHFND**: Normalized difference between LF and HF; **HR**: Heart rate; **RR**: Interval between R peaks. **DV (Dependent Variable).** The physiological measure being analyzed (e.g., LF, HF, LHFP, etc.). **Analysis.** Indicates whether the effect is (a) within-subject (e.g., Phase × AuthorityB) or (b) between-subject (e.g., Age, Gender, ClarityB). **F (df1, df2).** The F-statistic and relevant degrees of freedom. **p.** The exact p-value reported (or threshold if only given as "< 0.001"). **ηp².** Partial eta squared, a measure of effect size. **Notes.** Key interpretation points: whether the measure increased or decreased in each phase or group, pairwise comparisons (MD, mean difference), etc. All F-values, degrees of freedom (df1, df2), p-values, partial eta-squared (ηp²) values, and mean differences (MD) refer to the specific contrasts or effects reported in the manuscript. Some contrasts are reported only in text (e.g., "Within-Subjects Contrast") where the primary ANOVA yielded significance, post-hoc or planned contrasts further clarified the direction of the effect. Non-significant interactions or effects (e.g., p ≥ 0.05) are omitted for clarity.

Conversely, men produced a significantly higher number of publications than women, as well as higher h-index scores. The difference held even after controlling for age and field. This discrepancy highlights a gender gap in academic productivity metrics, which could be influenced by various factors, starting from the unbalanced parental investment that female and male scientists experience when having a child [45]. Another possible factor that might influence this difference is that outliers in productivity are mostly found among male scientists [46], although the logarithmic transformation applied to our data should have reduced this effect.

In many professional settings, women in public-facing roles report greater pressure to perform effectively, a phenomenon that can motivate them to adopt more proactive coping and stress-management strategies [47]. In our sample, significantly more women (56.3%) than men (12.5%) received formal public speaking training.

Training in anxiety-management techniques, such as breath control and cognitive reframing, can directly influence physiological responses by reducing sympathetic nervous system activity and enhancing parasympathetic rebound [48]. This physiological impact underscores the significance of coping strategies in managing stress, particularly in high-pressure environments like public communication.

Furthermore, research indicates that women are more likely to engage in behaviors that build social resources and seek interpersonal support when facing stress, aligning with the "tend-and-befriend" response model [49]. Such behaviors not only help mitigate stress but may also contribute to improved physiological outcomes, such as higher heart rate variability (HRV), which is associated with better emotional regulation and resilience.

This may suggest that female science communicators are more proactive in seeking formal training or support to enhance their communication skills, potentially as a compensatory strategy to overcome greater levels of emotionality, gender biases, and barriers they might face in the academic and public communication arenas. Alternatively, it could indicate a greater recognition among females of the importance of such training in enhancing their effectiveness and confidence as science communicators [50].

Overall, these psychological and gender-related findings provide a nuanced understanding of the attributes and challenges faced by science communicators. The high levels of self-esteem and prosociality observed are positive indicators of the effectiveness and resilience of these individuals in their roles [51].

### 4.2. *Public evaluation of science communication performance*

The external jury's examination of the six communication ability questionnaire dimensions yielded an exhaustive evaluation of the science communicators' performance. The mean scores across the dimensions (measured as Likert scale: 0-5) ranged from 3.27 (Interest in the Topic) to 3.92 (Clarity of Presentation), indicating generally positive evaluations of the communicators. This reflects a high level of competency among the science communicators in effectively presenting and engaging their audience.

The moderate to high correlations among the variables suggested the existence of one or more underlying dimensions measured by the questionnaire. PCA revealed a two-component solution that explained 75% of the variance, highlighting two distinct aspects of science communication: Content and Performance. The first component, Content, included Interest in the Topic and Agreement with the communicator. The second component, Performance, comprises Authoritativeness, Clarity of presentation, and Learning impact, emphasizing the importance of the communicator's delivery and ability to educate the audience effectively. Engagement loaded nearly equally on both components, suggesting it depends on both the content's appeal and the communicator's delivery style (Fig 4).

Correlational analyses provided further insights into the relationships between demographic and psychological variables with the interview judgments. Age was positively correlated with several dimensions, suggesting that older science communicators may bring more clarity, authority, and engagement to their presentations, potentially due to greater experience and expertise. Prosociality was significantly correlated with Engagement, Interest, and Agreement (Content component), probably indicating that communicators with higher prosocial tendencies are more likely to engage in research that resonates with the interests of the community. Self-esteem was also positively correlated with Agreement, suggesting that communicators with higher self-esteem may be perceived as more convincing [51]. Interestingly, prosociality and self-esteem were the only two personality variables that were higher than average in our science communicators sample, indicating an implicit selection of personality traits related to better engagement in this work. Professional work experience, such as years teaching at university, was positively correlated with Interest. Similarly, various forms of public engagement, including the number of printed interviews, live TV interviews, recorded TV interviews, and press communications, were correlated with both Interest and Agreement, underscoring the importance of global more than specific communication experiences in understanding how to align with the public sentiments.

Furthermore, the logarithm of the number of publications was negatively correlated with Authoritativeness, suggesting that a higher number of publications may not necessarily translate to higher perceived authority in public communication contexts. Some factors might explain this inverse relationship. First, researchers with a high volume of publications often possess highly specialized knowledge within a narrow field [52]. While this depth of

expertise is valuable within academic realms, it may not translate as effectively to broader public audiences who require more generalized and accessible explanations. Additionally, the skills required for successful academic publishing—such as technical writing and data analysis—do not necessarily overlap with those needed for effective public communication [53]. Public engagement demands storytelling, simplifying complex concepts, and engaging diverse audiences. Researchers focused on publishing may have had fewer opportunities to develop these distinct communication skills, impacting their perceived authoritativeness in public forums.

### 4.3. Psychophysiological profile of science communicators

The pairwise comparisons of physiological responses across different interview phases offer valuable insights into the stress dynamics that science communicators experience during live interviews. By adjusting for age and the logarithm of the number of publications as covariates, these comparisons revealed significant variations in HRV metrics, indicating distinct physiological states before, during, and after the interview. Specifically, the study found that total power, LHFP, LF, and SDNN components of HRV were significantly higher in the post-interview phase compared to both the interview and pre-interview phases. This increase suggests a recovery phase where the ANS returns to a more balanced state following the stress of the live interview, or potentially a rebound effect after the sharp decrease observed during the interview phase.

The behavior of LF during acute mental stress remains debated in the literature, with some studies reporting an increase and others a decrease in LF [54]. In our findings, a rapid decrease in LF may indicate heightened sympathetic activation, possibly due to a quick shift in the autonomic balance towards a predominantly sympathetic state, accompanied by parasympathetic inhibition [55]. Stressful situations can lead to significant changes in breathing patterns, which may suppress respiratory sinus arrhythmia (RSA)—a major contributor to both high-frequency (HF) and low-frequency (LF) components of HRV. Furthermore, when stress responses reach saturation, both LF and HF band magnitudes may decrease [56]. During cognitively and emotionally demanding tasks, such as a live interview, the increased mental and emotional load can divert attentional resources away from autonomic regulation, resulting in a decrease in HRV, including LF.

SDNN's behavior provides further insights into the ability of the ANS to dynamically adjust to stressors and recover effectively. SDNN is widely recognized as an index of physiological resilience to stress, with a decrease in SDNN indicating poor autonomic function [33] The observed increase in SDNN during the post-interview phase suggests a recovery period following the acute stress of live interviews. This indicates that participants' ANS was able to adapt effectively, which is critical for high-performance communicators to maintain focus, composure, and resilience under stress.

Total power, as a comprehensive measure of overall HRV, reflects the total variance of RR intervals and reflects the overall capacity of the ANS to respond to and recover from stress [32]. According to the Task Force's guidelines, sympathetic activation typically results in tachycardia accompanied by a marked reduction in total power [33]. Therefore, the observed increase in total power during the post-interview phase aligns with expectations. This finding suggests that the participants were transitioning into a recovery state, where a robust ANS response facilitates the restoration of physiological balance following stress.

HF components, closely associated with PNS activity, were significantly higher in the post-interview phase compared to the interview phase. This finding aligns with a recent meta-analysis showing that acute psychological stress is characterized by low parasympathetic activity, as evidenced by decreases in the HF band [57].

Additionally, RR intervals were significantly longer, and HR significantly lower, during both the post- and pre-interview phases compared to the interview phase. These findings suggest slower HRV intervals and a more stable, relaxed physiological state before and after the interview, in contrast to the heightened stress response observed during the interview itself [58].

When analyzing the data by gender, both males and females exhibited similar overall patterns in physiological responses across the different phases. However, significant differences in total power, LHFP, LF, and HF components across phases were observed only in the total sample and for female participants. Female participants also had overall higher RR intervals and lower HR ranges compared to males. These results suggest that female researchers might maintain a more stable and relaxed physiological state throughout the interview process but might experience a more pronounced difference in stress response than their male counterparts [59].

Furthermore, these differences may stem from distinct hormonal profiles. For instance, estrogen has been associated with an enhanced vagal tone and better parasympathetic reactivity, potentially leading to higher HRV in women under stress or during recovery periods [60]. Indeed, a meta-analysis examining sex differences in HRV found that women tend to have higher parasympathetic activity than men, which in turn may facilitate quicker autonomic recovery after acute stressors [61].

Additionally, it is worth noting that women may show different autonomic reactivity across the menstrual cycle [60], such that the phase of the cycle or menopause could modulate stress responses through estrogen- and progesterone-related effects on the autonomic nervous system. Although we did not collect menstrual cycle data in this study, future work could explore how these hormonal fluctuations contribute to real-time stress regulation in communication contexts.

As discussed in the previous section, women's elevated HRV may reflect the combined effect of physiological predispositions and enhanced behavioral strategies for coping with on-camera stress.

Taken together, these findings underscore the importance of considering both physiological (e.g., hormonal) and psychosocial (e.g., training, cultural expectations) factors when explaining gender differences in stress responses and HRV profiles. By examining these dimensions in tandem, it is possible to better understand how expertise in public speaking is acquired, manifested, and modulated across diverse communicator populations.

### 4.4. Comparing high and low performance on bidimensional physiological responses

Communicators were divided into high (HA) and low Authoritativeness (LA) and high (HC) and low Clarity (LC) according to cluster analysis. This dichotomic division allowed us to better address the effects of performance on physiological responses. An analysis using Repeated Measures ANOVA and grouping participants based on the jury judgments (Authoritativeness; Clarity) further revealed an interaction between authority, clarity, and HRV-related parameters.

The analysis revealed that both clarity and authority play significant roles in modulating LF power across the different phases of the interview process. For individuals perceived as having lower authority or lower clarity, LF power exhibited greater fluctuations—decreasing during the interview and rebounding post-interview. This pattern suggests that these individuals experience more pronounced sympathetic reactivity when under stress, leading to a heightened recovery phase once the stressor is removed. In contrast, those with higher authority and

higher clarity demonstrated more stable LF power throughout the phases, indicating a more consistent autonomic response and potentially better stress management. HF power, on the other hand, did not show significant phase-related effects in relation to authority. However, the interaction between phase and clarity was notable, with clearer communicators showing an increase in HF power during the post-interview phase. This suggests that clarity in communication may facilitate better parasympathetic recovery after stress, helping individuals to calm down more effectively after the high-pressure situation of an interview. When considering HF and LF together, the patterns observed indicate that individuals with higher clarity and authority are better able to maintain a balance between sympathetic activation and parasympathetic recovery.

In this analysis, LHFP and LHFND were used to characterize bidimensional autonomic responses across different performance groups. LHFP provides an overall measure of autonomic power by combining LF and HF components, while LHFND offers a ratio that reflects the balance between these two components, with higher values typically indicating greater sympathetic dominance. Unlike the LF/HF ratio, which can be distorted by extreme values which can occur when either LF or HF approaches zero, LHFND provides a symmetric, bounded index that is more robust and interpretable in dynamic contexts. The analysis of LHFP showed similar patterns to those observed with LF power. In the LA and LC groups, LHFP significantly decreased during the interview phase and then increased post-interview. This suggests that these individuals experience greater autonomic strain during the interview, reflected in reduced overall autonomic power, followed by a pronounced recovery phase. In contrast, the HA and HC groups maintained more stable LHFP across all phases, indicating a more balanced autonomic response and a similar state maintained through all the phases. LHFND analysis further supported these findings, highlighting a more balanced autonomic response in the HA group compared to the LA group. This balance was less pronounced in the LC group when compared to those with higher clarity, underscoring the utility of LHFND as a refined metric for capturing autonomic balance in high-stress scenarios and differentiating adaptive responses between performance groups.

### 4.5. HRV indices as a new biomarker of science communication expertise

Implementing a two-dimensional analysis (HF vs. LF) of HRV offers several advantages over the traditional use of the LF/HF ratio [56], providing a more nuanced understanding of ANS dynamics. By analyzing these components separately, we were able to gain clearer insights into the specific contributions of sympathetic and parasympathetic regulation. This distinction is important for the identification of which part of the ANS is more active [52] and allows us to understand how communicators manage stress during different phases of public speaking.

In the HF-LF plan, the upper-left quadrant (high-LF, low-HF) is characteristic of high physical/low mental stress, the lower-left quadrant (low-LF, low-HF) of low physical/high emotional stress, while, for mental stress, HF is low, but data on LF is inconsistent [54,56]. The upper-right quadrant (high-LF, high-HF) can be interpreted as the first phase of recovery/rebound as the person experiences mental fatigue [55] but starts recovering the fastest HF resting values. Usually a balanced LF-HF (mid-right) value is found in the resting state [54,56], while active engagement or flow state shows a moderate decrease in HF compared to rest, while maintaining moderate levels of LF (mid-left) [62].

### 4.6. Limitations

Some limitations should be acknowledged. First, the sample size of thirty-two participants, further reduced to 25 for HRV analysis, although adequate for preliminary analysis, limits the

generalizability of the findings. A larger and more diverse sample would provide more robust data and improve the external validity of the results. The sample was also at the high end of their career and concentrated in a specific institution and geographical location. This limitation may affect the generalizability of the findings to science communicators from different cultural backgrounds and career stages. Cultural factors are another factor that might affect our data. Indeed, it has been demonstrated that science communication styles may vary by culture [63]. Research comparing college students in the US and East Asia shows that fundamental attentional and perceptual processes differ between cultures. For instance, East Asians focus more on relationships and background information (context) than US participants, who are more prone to focus on fore-grounded, focal items [64].

Furthermore, the jury that used to evaluate the effectiveness of the science communicators was composed predominantly by young undergraduate students. Even though we acknowledge that this limits the representativeness of the population, we shall also emphasize that young adults are typically the ones who score higher in literacy [58], and recent findings show that they are also better at objectively comprehending pandemic-related scientific communication [65]. However, to provide a more accurate estimate of scientific communication quality, future studies may make use of peer evaluations from peer science communicators.

While gender differences were explored, the sample might not be representative of the broader population of science communicators in terms of gender distribution, especially across disciplines. This could influence the interpretation of gender-specific findings and their applicability to a wider context.

Finally, the study was conducted in a naturalistic but specific environment which may not fully replicate the wide range of stressors encountered in more varied real-world public speaking scenarios. The ecological validity could be enhanced by including a broader array of public engagement settings.

## 5. Conclusions

We provided for the first time the psychophysiological correlates of expert science communicators. The significant physiological changes observed across different interview phases highlight the dynamic nature of stress and recovery, emphasizing the need for effective stress management strategies and the importance of experience. Perceived authority and clarity are associated with mitigating stress responses measured through HRV. Tailored training programs focusing on building authority and confidence could be particularly beneficial for science communicators, helping them manage stress more effectively. Nowadays, in university or school explicit training in non-expert communicators is still unusual because the focus is usually on developing scientist-to-scientist communication abilities through peer-reviewed publications and conference presentations. Scientists who are reluctant to engage in public activities sometimes claim a lack of training or confidence in their ability to communicate science [6]. For this reason, several rules, guidelines have been published for laying the foundation of the best practice for scientific communication. Despite there being several excellent courses for training effective communication skills, none lies on the psychophysiological control of ANS. Evidence provided in this study could be transferred to specific training programs where non-expert communicators could learn to control mental stress by training in HRV-related modulatory changes during other types of science communication settings (i.e., social media or scientific conferences) using a biofeedback approach. The ultimate goal is to prepare the future generation of scientists to deal with challenging communication situations, like audiences that are skeptics or divided (see the COVID-19 pandemic crisis [66]).

## Supporting information

**S1 File. This is a summplementary document with additional data analysis.**
(DOCX)

**S1 Table. This is an additional table reporting supplementary data about external injury scores.**
(XLSX)

**S2 Table. This is an additional table reporting all data recorded our experiment.**
(XLSX)

## Author contributions

**Conceptualization:** Gennaro Tartarisco, Antonio Cerasa.

**Data curation:** Gennaro Tartarisco, Simona Campisi, Loredana Cerbara, Marco Dedola, Francesca Gorini.

**Formal analysis:** David Vagni, Antonio Tintori.

**Investigation:** David Vagni, Gennaro Tartarisco, Marco Dedola, Alessandra Pedranghelu, Alexandra Castello, Chiara Failla.

**Methodology:** David Vagni, Antonio Tintori.

**Project administration:** Marco Tullio Liuzza, Giovanni Pioggia, Marco Ferrazzoli.

**Resources:** Marco Tullio Liuzza.

**Supervision:** Antonio Cerasa.

**Writing – review & editing:** David Vagni, Marco Tullio Liuzza, Antonio Cerasa.

## Acknowledgments

We would like to thank Andrea Bettini, Director of the Rainews24, for inviting us to participate in this psychosocial study. We would thank Alessandro Vecchi, Luna Giada Mazza, Melina Perri, and Matilde Lico for their help in collecting data. Roberta Bruschetta is a PhD student enrolled in the National PhD in Artificial Intelligence, XXXVII cycle, course on Health and life sciences, organized by Università Campus Bio-Medico di Roma; Simo- na Campisi is a PhD student enrolled in the National PhD in Artificial Intelligence, XXXVIII cycle, course on Health and life sciences, organized by Università Campus Bio-Medico di Roma.

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
