## [Decision Letter · Decision Letter 0]

13 Dec 2024

Dear Dr. Cerasa,

We look forward to receiving your revised manuscript.

Kind regards,

Rasool Abedanzadeh, Ph.D

Academic Editor

PLOS ONE

Journal Requirements:

2. We note that Figure 1 in your submission contain copyrighted images. All PLOS content is published under the Creative Commons Attribution License (CC BY 4.0), which means that the manuscript, images, and Supporting Information files will be freely available online, and any third party is permitted to access, download, copy, distribute, and use these materials in any way, even commercially, with proper attribution. For more information, see our copyright guidelines: http://journals.plos.org/plosone/s/licenses-and-copyright .

Reviewers' comments:

Reviewer's Responses to Questions

**Comments to the Author**

1. Is the manuscript technically sound, and do the data support the conclusions?

Reviewer #1: Partly

2. Has the statistical analysis been performed appropriately and rigorously?

Reviewer #1: N/A

3. Have the authors made all data underlying the findings in their manuscript fully available?

Reviewer #1: Yes

4. Is the manuscript presented in an intelligible fashion and written in standard English?

Reviewer #1: Yes

Reviewer #1: General Comment:

The manuscript, "Psychophysiological correlates of science communicators," addresses a novel and important topic by examining how science communicators' physiological responses—specifically heart rate variability (HRV)—relate to their communication skills during high-stress situations like live television interviews. The study's ecological design is a strength, and the findings provide useful insights into the intersection of stress management and effective science communication. However, several areas could benefit from deeper analysis and clarification, particularly regarding the physiological measures, gender-related differences, and methodological approaches.

Specific Comments:

HRV Analysis and Physiological Insights: The use of HRV as a measure of autonomic nervous system regulation is appropriate, but the interpretation of HRV metrics (LF, HF, SDNN) in the context of stress and performance requires further detail. Specifically:

Low-frequency (LF) and high-frequency (HF) components are often debated in terms of their association with sympathetic and parasympathetic activity. The manuscript presents LF as a measure of sympathetic and parasympathetic activity and HF as predominantly parasympathetic, but this simplification can be misleading. More discussion is needed on the controversial nature of the LF/HF ratio as a marker of sympathovagal balance, including the challenges raised in the literature (e.g., Billman, 2013; von Rosenberg et al., 2017).

The manuscript would benefit from expanding the explanation of how SDNN (Standard Deviation of NN intervals) relates to overall autonomic flexibility and stress resilience, particularly in high-performance communicators. The role of total power and its relationship to stress recovery and resilience could be elaborated to connect better with the broader literature on HRV and stress response.

While the two-dimensional analysis of HRV (HF and LF components) is innovative, the rationale behind introducing the Low-High Frequency Normalized Difference (LHFND) index should be clarified. What advantages does LHFND offer over traditional metrics, and how does it better capture the nuances of autonomic balance in this context? Providing real-world examples or literature references where this approach has been beneficial could strengthen the argument.

Gender Differences in HRV and Communication: The gender differences in HRV responses, particularly in RR intervals, total power, and LF/HF ratios, are notable. However, the manuscript does not sufficiently address the physiological basis for these differences, nor does it critically examine the broader implications of these findings. Suggestions for improvement include:

A deeper exploration of why female communicators exhibited higher HRV and lower heart rate ranges. This could be tied to literature on gender-specific responses to stress, as well as possible hormonal influences (e.g., differences in parasympathetic tone related to estrogen).

While the manuscript attributes the emotionality scores to gender, it could benefit from a discussion of social and cultural factors that might influence these responses. For instance, women in public-facing roles might experience additional pressure to perform well, which could influence their physiological stress responses.

There is a mention that women received more public speaking training than men, which might explain part of their better performance in some HRV parameters. The manuscript should expand on how this training might influence stress regulation and whether this reflects a gendered coping strategy in high-stress communication contexts.

Jury-Based Evaluation Methodology: The evaluation of science communicators by a jury of university students introduces potential biases, especially given the age, experience, and expertise differences between the evaluators and the communicators. To strengthen the credibility of the evaluation method:

Discuss the potential limitations of using a relatively inexperienced audience to assess expert science communicators. How might the evaluators' lack of experience influence their perception of clarity, authoritativeness, and engagement? Would more seasoned professionals provide different evaluations?

Consider discussing alternative or additional methods for evaluating communication skills, such as peer review by fellow science communicators or feedback from the broader public. Comparing the results from different types of audiences could provide a richer understanding of how communication effectiveness varies across contexts.

The Principal Component Analysis (PCA) revealed two main dimensions—content and performance. It would be helpful to explain whether these components are consistent with findings from previous research on science communication, and how these dimensions might inform future training programs for communicators.

Statistical Analysis and Presentation: While the manuscript provides thorough statistical analysis, some sections could be more accessible to readers unfamiliar with advanced statistical methods:

The multivariate analyses and repeated measures ANOVA results are crucial, but the interaction effects between communication performance (authoritativeness, clarity) and HRV components would benefit from clearer visual representation. For example, a summary table or diagram illustrating the key interaction effects (e.g., between authority levels and HRV changes across phases) could make the results more digestible.

The manuscript uses Box’s Test of Equality of Covariance Matrices and Mauchly's Test of Sphericity but does not explain their implications for the reader. Providing a brief explanation of why these tests are relevant (e.g., testing assumptions for ANOVA) would improve the statistical transparency of the study.

Broader Implications and Applications: The findings have potential practical applications, especially for developing training programs for science communicators. However, the manuscript could elaborate on how these results could be translated into real-world practice:

Training for Science Communicators: The study suggests that perceived authoritativeness and clarity help reduce stress responses. How might training programs integrate biofeedback techniques to help communicators manage their autonomic responses during high-stress events? Discuss the potential for HRV biofeedback to become part of standard training for public speakers.

Future Research Directions: The study opens the door to many avenues for further investigation. How might these findings apply to other types of science communication settings (e.g., social media or scientific conferences)? Additionally, how do communicators’ HRV responses differ when engaging with more ideologically polarized or skeptical audiences, as these are common challenges in science communication?

Limitations: The limitations section touches on key aspects like sample size and the institutional context, but it could be expanded to discuss other important limitations:

Cultural factors: The study focused on Italian communicators. Science communication styles and stress responses may vary by culture, so the generalizability of the findings might be limited. Addressing this and suggesting cross-cultural studies could strengthen the manuscript.

Ecological Validity: While the live television setting is a strength, it may not fully capture the range of stressors experienced in other science communication scenarios, such as hostile public debates or high-pressure academic presentations. Expanding on this point and proposing ways to test these findings in different environments would provide more comprehensive insight.

Conclusion:

This manuscript presents important and original findings on the psychophysiological correlates of science communication expertise. The data on HRV in relation to performance skills like clarity and authoritativeness are insightful and could have significant implications for the training of science communicators. However, the manuscript would benefit from deeper analysis of the HRV metrics, clearer presentation of the statistical interactions, and a more nuanced discussion of gender differences. By addressing these areas, the manuscript could offer a more comprehensive and impactful contribution to both the fields of psychophysiology and science communication.

**Do you want your identity to be public for this peer review?** For information about this choice, including consent withdrawal, please see our Privacy Policy

Reviewer #1: **Yes: ** professor Sofia Pavanello

---

## [Author Response · Author response to Decision Letter 0]

8 Jan 2025

Dear Editor,

We express our gratitude to the reviewer for their insightful and stimulating comments. We appreciate the fact that this reviewer considers our paper as “a more comprehensive and impactful contribution to both the fields of psychophysiology and science communication”

The list of modifications made now to our manuscript has been highlighted in yellow.

To Reviewer 1:

The manuscript, "Psychophysiological correlates of science communicators," addresses a novel and important topic by examining how science communicators' physiological responses—specifically heart rate variability (HRV)—relate to their communication skills during high-stress situations like live television interviews. The study's ecological design is a strength, and the findings provide useful insights into the intersection of stress management and effective science communication. However, several areas could benefit from deeper analysis and clarification, particularly regarding the physiological measures, gender-related differences, and methodological approaches.

Specific Comments:

1) HRV Analysis and Physiological Insights: The use of HRV as a measure of autonomic nervous system regulation is appropriate, but the interpretation of HRV metrics (LF, HF, SDNN) in the context of stress and performance requires further detail. Specifically:

a. Low-frequency (LF) and high-frequency (HF) components are often debated in terms of their association with sympathetic and parasympathetic activity. The manuscript presents LF as a measure of sympathetic and parasympathetic activity and HF as predominantly parasympathetic, but this simplification can be misleading. More discussion is needed on the controversial nature of the LF/HF ratio as a marker of sympathovagal balance, including the challenges raised in the literature (e.g., Billman, 2013; von Rosenberg et al., 2017).

REPLY: Thank you for your valuable feedback regarding the interpretation of HRV metrics and the LF/HF ratio. We have revised the manuscript to incorporate a finer discussion of these components, addressing the controversies associated with their physiological interpretation. Specifically:

LF Component: We have clarified in the revised manuscript that LF power reflects a combination of both sympathetic and parasympathetic activity, with a significant contribution from baroreflex-mediated autonomic modulation, as supported by Billman (2013) and other works.

HF Component: HF power remains characterized as a marker of PNS activity, particularly RSA. We have highlighted that stress-induced alterations in respiration can impact both HF and LF components, aligning with existing literature.

In response to the concerns about the LF/HF ratio, we expanded the discussion to address its limitations, including its susceptibility to distortions due to non-linear relationships and extreme values. To overcome these limitations, our analysis focuses on separate evaluations of LF and HF components and introduces a novel index, the LHFND.

b. The manuscript would benefit from expanding the explanation of how SDNN (Standard Deviation of NN intervals) relates to overall autonomic flexibility and stress resilience, particularly in high-performance communicators. The role of total power and its relationship to stress recovery and resilience could be elaborated to connect better with the broader literature on HRV and stress response.

REPLY: We appreciate the reviewer’s comment and agree that a more detailed discussion of SDNN and Total Power’s roles in stress resilience and recovery would enhance the manuscript. In response, we have expanded Section 4.3 of the Discussion to adequately address these aspects.

Regarding SDNN, we emphasized its role as an indicator of overall HRV, reflecting the total variability of the autonomic nervous system (ANS) by integrating both PNS and SNS inputs. Increased SDNN has been linked to greater physiological resilience and the capacity of the ANS to dynamically adapt to stressors and recover effectively. For high-performance communicators, such adaptability is essential for maintaining focus, composure, and stress resilience during live interviews. The observed increase in SDNN during the post-interview phase aligns with this interpretation, indicating a recovery period during which participants transitioned to a more balanced autonomic state after the acute stress experienced during the interview.

For total power, we elaborated on its role as a measure of overall HRV, reflecting the total variance of RR intervals and the overall capacity of the ANS to respond to and recover from stress. Additionally, we linked our findings to the Task Force guidelines, which indicate that sympathetic activation often leads to reduced total power. The observed increase in total power during the post-interview phase aligns with this aspect and supports the interpretation of a recovery phase following the stress of the interview.

c) While the two-dimensional analysis of HRV (HF and LF components) is innovative, the rationale behind introducing the Low-High Frequency Normalized Difference (LHFND) index should be clarified. What advantages does LHFND offer over traditional metrics, and how does it better capture the nuances of autonomic balance in this context? Providing real-world examples or literature references where this approach has been beneficial could strengthen the argument.

REPLY: To clarify the rationale and advantages of the LHFND index we have expanded the explanation in Section 2.7 and the Discussion (Section 4.4) to better articulate the motivation for using LHFND and its interpretive value. Specifically, the index offers several advantages over traditional metrics like the LF/HF ratio:

Unlike LF/HF, LHFND is bounded symmetrically between -1 and +1, providing a more intuitive interpretation of automatic balance, with values near -1 indicating HF dominance, +1 indicating LF dominance, and 0 representing a balanced state.

The LF/HF ratio can become distorted when LF or HF values approach zero, leading to extreme and less interpretable results. LHFND avoids such distortions by normalizing the difference between LF and HF.

LHFND builds on the normalized difference approach widely used in other fields, as specified in the manuscript. Applying this approach to HRV allows us to capture meaningful shifts in autonomic dynamics.

The integration of LHFND with LHFP in a two-dimensional analysis further strengthens its utility, as it combines relative autonomic balance (LHFND) with overall autonomic power (LHFP).

2) Gender Differences in HRV and Communication: The gender differences in HRV responses, particularly in RR intervals, total power, and LF/HF ratios, are notable. However, the manuscript does not sufficiently address the physiological basis for these differences, nor does it critically examine the broader implications of these findings. Suggestions for improvement include:

a. A deeper exploration of why female communicators exhibited higher HRV and lower heart rate ranges. This could be tied to literature on gender-specific responses to stress, as well as possible hormonal influences (e.g., differences in parasympathetic tone related to estrogen).

REPLY: We agree that our initial discussion on gender differences in HRV could benefit from further elaboration on the underlying physiological mechanisms. There is evidence suggesting that female communicators might exhibit higher parasympathetic tone (reflected in higher HRV and lower heart rate range) possibly due to factors such as hormonal influences (e.g., estrogen levels) and their role in modulating vagal tone. Additionally, some studies have shown that women may exhibit distinct autonomic responses to stress, displaying stronger parasympathetic reactivation once an acute stressor has ended. We will add a section in the Discussion to incorporate references that link female HRV responses to potential hormonal pathways and to broader findings about gender-specific stress reactivity (see pag. 34).

These physiological patterns may also interact with psychological and socio-cultural variables. For instance, higher levels of emotionality (as measured by self-report scales) could amplify the visibility of these autonomic differences; some women who score higher in emotionality might also be more adept at recruiting coping mechanisms—both conscious (e.g., breathing techniques) and unconscious (physiological resilience)—resulting in higher HRV values.

b. While the manuscript attributes the emotionality scores to gender, it could benefit from a discussion of social and cultural factors that might influence these responses. For instance, women in public-facing roles might experience additional pressure to perform well, which could influence their physiological stress responses.

There is a mention that women received more public speaking training than men, which might explain part of their better performance in some HRV parameters. The manuscript should expand on how this training might influence stress regulation and whether this reflects a gendered coping strategy in high-stress communication contexts.

REPLY: We appreciate this recommendation and agree that social and cultural factors can significantly influence gender differences in stress responses. The finding that more women than men received formal public speaking training in our study aligns with literature suggesting women in public-facing or leadership roles often seek additional skill-building opportunities to bolster confidence and mitigate performance anxiety.

We will expand the Discussion by highlighting how these trainings might confer better stress-regulation strategies, thereby manifesting in higher HRV and lower heart rate ranges under pressure. Moreover, cultural norms and expectations may place extra pressure on women to communicate effectively and project confidence in public settings—this could intensify the motivation to develop coping strategies or might itself alter physiological stress responses (e.g., “tend-and-befriend” patterns, as proposed in some stress models). In our revision, we will situate our findings in the broader context of these gendered coping strategies, referencing additional work on how socio-cultural expectations shape women’s physiological and psychological stress responses.

3) Jury-Based Evaluation Methodology: The evaluation of science communicators by a jury of university students introduces potential biases, especially given the age, experience, and expertise differences between the evaluators and the communicators. To strengthen the credibility of the evaluation method:

a. Discuss the potential limitations of using a relatively inexperienced audience to assess expert science communicators. How might the evaluators' lack of experience influence their perception of clarity, authoritativeness, and engagement? Would more seasoned professionals provide different evaluations?

REPLY:The evaluation served the purpose to provide some criterion validity about how effective the science communicators were. It was not meant to prove an accurate estimate of their real competence, therefore we do not think that their lack of experience could affect the results. If anything, we may assume that a younger population could be more competent about the topics covered in the shows, as the OCSE PIAAC 2024 (https://doi.org/10.1787/b263dc5d-en) data show that people in the younger end of the Italian population are better in literacy, numeracy and problem solving. Also, research on scientific communication reception showed that young people objectively comprehend better (Lungu et al., 2024) pandemic-related videos. However, we capitalized in this observation adding the following part to the limitation section in the discussion:

“The jury that used to evaluate the effectiveness of the science communicators was composed predominantly by young undergraduate students. Even though we acknowledge that this limits the representativeness of the population, we shall also emphasize that young adults are typically the ones who score higher in literacy (OECD, 2024), and recent findings show that they are also better at objectively comprehending pandemic-related scientific communication (Lungu et al., 2024).”

b. Consider discussing alternative or additional methods for evaluating communication skills, such as peer review by fellow science communicators or feedback from the broader public. Comparing the results from different types of audiences could provide a richer understanding of how communication effectiveness varies across contexts.

REPLY: We thank the Reviewer for this observation, we added these considerations as a potential future direction in the limitations section: “However, to provide a more accurate estimate of scientific communication quality, future studies may make use of peer evaluations from peer science communicators. “

c. The Principal Component Analysis (PCA) revealed two main dimensions—content and performance. It would be helpful to explain whether these components are consistent with findings from previous research on science communication, and how these dimensions might inform future training programs for communicators.

REPLY: To the best of our knowledge no similar ecological psychophysiological studies have never been conducted on science communicators. As a result, we lack the elements we need to discuss how content and authoritativeness are associated with communication performance. However we now include a new section discussing future directions in this new field of study.

4) Statistical Analysis and Presentation: While the manuscript provides thorough statistical analysis, some sections could be more accessible to readers unfamiliar with advanced statistical methods:

a. The multivariate analyses and repeated measures ANOVA results are crucial, but the interaction effects between communication performance (authoritativeness, clarity) and HRV components would benefit from clearer visual representation. For example, a summary table or diagram illustrating the key interaction effects (e.g., between authority levels and HRV changes across phases) could make the results more digestible.

REPLY: We recognize that the current manuscript’s detailed statistical presentation may benefit from more straightforward visual aids. In the revised version, we will include three additional tables (Table 4,5,6) that consolidate the key interaction effects between communication performance (e.g., Authoritativeness, Clarity) and HRV changes across the three phases (pre-interview, interview, post-interview). Those tables will highlight the differences between High vs. Low Authoritativeness and High vs. Low Clarity groups in a user-friendly format (Table 4, Table 5 and Table 6).

b. The manuscript uses Box’s Test of Equality of Covariance Matrices and Mauchly's Test of Sphericity but does not explain their implications for the reader. Providing a brief explanation of why these tests are relevant (e.g., testing assumptions for ANOVA) would improve the statistical transparency of the study.

REPLY: We appreciate the suggestion to clarify why we conducted these tests. In our revision, we will add a brief explanation in the Statistical Analysis section: Box’s Test of Equality of Covariance Matrices checks whether the covariance matrices are equal across groups—a key assumption in multivariate analyses (e.g., MANOVA). If the test is significant, it indicates that the assumption of homogeneity of covariances is violated and alternative analytical approaches or corrections might be needed. Mauchly’s Test of Sphericity is relevant when using repeated-measures ANOVA and assesses whether the variances of the differences between all possible pairs of within-subject conditions are equal. Violation of sphericity can inflate Type I error rates, so when it is significant, we apply corrections such as the Greenhouse-Geisser or Huynh-Feldt adjustments.

5) Broader Implications and Applications: The findings have potential practical applications, especially for developing training programs for science communicators. However, the manuscript could elaborate on how these results could be translated into real-world practice:

a. Training for Science Comm

---

## [Decision Letter · Decision Letter 1]

14 Feb 2025

Psychophysiological correlates of science communicators.

PONE-D-24-40963R1

Dear Dr. Cerasa,

We’re pleased to inform you that your manuscript has been judged scientifically suitable for publication and will be formally accepted for publication once it meets all outstanding technical requirements.

Kind regards,

Rasool Abedanzadeh, Ph.D

Academic Editor

PLOS ONE

Additional Editor Comments (optional):

Reviewers' comments:

Reviewer's Responses to Questions

**Comments to the Author**

Reviewer #1: All comments have been addressed

2. Is the manuscript technically sound, and do the data support the conclusions?

Reviewer #1: Yes

3. Has the statistical analysis been performed appropriately and rigorously?

Reviewer #1: Yes

4. Have the authors made all data underlying the findings in their manuscript fully available?

Reviewer #1: Yes

5. Is the manuscript presented in an intelligible fashion and written in standard English?

Reviewer #1: Yes

Reviewer #1: Evaluation of Revised Manuscript PONE-D-24-40963R1

I appreciate the authors' thorough and thoughtful revisions, which have significantly improved the manuscript. They addressed all major concerns raised in the initial review:

HRV Metrics and LF/HF Ratio: The discussion has been expanded to include the controversies surrounding the LF/HF ratio, with the introduction of the LHFND index providing a more robust and interpretable approach to analyzing autonomic balance.

Role of SDNN and Total Power: The authors have effectively connected these metrics to stress resilience and recovery, citing relevant guidelines and literature.

Gender Differences: The physiological and socio-cultural bases for observed gender differences in HRV were well-explored, enhancing the manuscript's depth.

Jury Methodology: The limitations of using a student jury were acknowledged, and alternative approaches for future studies were suggested.

Statistical Analysis: The inclusion of additional tables and clear explanations has greatly improved the accessibility and transparency of the statistical findings.

Practical Implications: The discussion on potential applications of HRV biofeedback in science communication training is highly relevant and forward-looking.

The manuscript now offers a more comprehensive and impactful contribution to the fields of psychophysiology and science communication. I recommend accepting the revised manuscript for publication.

**Do you want your identity to be public for this peer review?** For information about this choice, including consent withdrawal, please see our Privacy Policy

Reviewer #1: **Yes: ** Sofia Pavanello

---

## [Editor Report · Acceptance letter]

PONE-D-24-40963R1

PLOS ONE

Dear Dr. Cerasa,

I'm pleased to inform you that your manuscript has been deemed suitable for publication in PLOS ONE. Congratulations! Your manuscript is now being handed over to our production team.

Kind regards,

on behalf of

Dr. Rasool Abedanzadeh

Academic Editor

PLOS ONE